# GLYCO-BUILD: an enzymatic pipeline for the synthesis of peptides carrying eukaryotic *N*-glycans

Lorenzo Rossi [1,3], J. Andrew N. Alexander [1,3], Ana S. Ramírez [1,2,3] &
Kaspar P. Locher [1] ✉

Peptides and proteins containing defined *N*-glycans hold great promise for biomedicine because the attached glycans engage in specific interactions, influence targeted delivery and folding, and improve pharmacokinetic properties. However, challenges associated with producing homogeneous *N*-glycoconjugates hinder their wider application. Here, we report the development of GLYCO-BUILD, an enzymatic pipeline that can generate glycopeptides carrying eukaryotic *N*-glycans at a level of homogeneity not accessible by alternative methodologies. The pipeline recapitulates the endoplasmic reticulum-based steps of the eukaryotic protein *N*-glycosylation machinery. However, it employs a combination of enzymes of archaeal, bacterial, and eukaryotic origin that are able to process inexpensive phytol instead of the native dolichol as a lipid carrier. GLYCO-BUILD facilitates the synthesis of homogeneous *N*-glycans ranging from $GlcNAc_2$ to $GlcNAc_2Man_9Glc_3$, and their transfer to acceptor polypeptides using single-subunit oligosaccharyltransferases (OSTs) from the eukaryotic parasite *Trypanosoma brucei*. We used GLYCO-BUILD to generate glycopeptides mimicking viral mannosylated antigens, glucosylated species and precursors of complex and hybrid glycans. Our pipeline is modular, versatile and can be combined with other approaches of glycan extension and modification to generate a wide range of homogeneous *N*-glycoconjugates for use in research, diagnostics and therapeutics, including serum testing, vaccine development or modulation of biotherapeutics' half-life.

Protein *N*-glycosylation is an essential post-translational modification in eukaryotes. It is found in a majority of secretory proteins and consists of the attachment of specific oligosaccharides to asparagine residues that are part of glycosylation sequons (sequence N-X-S/T, where X can be any amino acid except proline)[1]. The process begins at the membrane of the endoplasmic reticulum (ER), where the conserved tetradecasaccharide $GlcNAc_2Man_9Glc_3$ is assembled onto the lipid carrier dolichylpyrophosphate, from where it is subsequently transferred to nascent polypeptides. The attached *N*-glycans have key

roles in protein folding and quality control in the ER, and facilitate export from the ER. As secretory glycoproteins pass through the Golgi apparatus, the originally attached, high-mannose *N*-glycan is remodeled, leading to a wide diversity of structures in the eukaryotic *N*-glycome. These are essential for multicellular life, as they mediate a broad range of cellular and organismal functions[2,3].

Beyond their physiological roles, *N*-glycans profoundly influence the pharmacological properties of peptide-based biotherapeutics[4,5]. They modulate peptide hydrophilicity, circulation time, and cellular

[1]Institute of Molecular Biology and Biophysics, Eidgenössische Technische Hochschule (ETH), Zürich, Switzerland. [2]Present address: Complex Carbohydrate Research Center, University of Georgia, Athens, GA, USA. [3]These authors contributed equally: Lorenzo Rossi, J. Andrew N. Alexander, Ana S. Ramírez. ✉e-mail: locher@mol.biol.ethz.ch

uptake, while simultaneously reducing aggregation, proteolysis and kidney clearance[6]. Attachment of homogeneous N-glycans represents a promising avenue to fine-tune the half-life and bioavailability of these therapeutics[7,8]. In addition, eukaryotic N-glycans decorate the surface proteins of many viruses, including HIV, Dengue and Hepatitis C[9–13]. Many of these structures are oligomannose glycans, which shield the existing protein epitopes and promote evasion from the host immune system. Synthetic glycopeptides carrying such structures are emerging as valuable tools for mimicking viral glycoantigens in immunological investigations[14–16].

Although their demand is steadily growing, the production of defined eukaryotic N-glycans and N-glycopeptides has remained challenging. Glycoconjugates representing intermediates or end products of the ER glycosylation pathways cannot be enriched or purified in large amounts from natural sources[3]. Glycoconjugates carrying complex or hybrid structures are also challenging to obtain as homogeneous species, since cellular N-glycan maturation is not template-driven[3]. At present, several approaches are employed for producing macromolecules carrying eukaryotic N-glycans, but each is associated with specific limitations. For the production of glycoproteins, overexpression in eukaryotic cells remains the most commonly employed method. However, this generally gives rise to heterogeneous N-glycans, which might require additional glycan remodeling in vitro[17–21]. Glycoprotein synthesis has also been attempted in engineered bacterial cells or in cell-free systems, but the use of bacterial oligosaccharyltransferases results in moderate efficiency and low yield[22–24]. Also, bacteria are unable to fold many therapeutically interesting eukaryotic proteins. For polypeptides up to a length of ~40 amino acids, solid-phase peptide synthesis can be used to incorporate amino acids with single sugars attached (for example, asparagine residues containing N-acetylglucosamine (GlcNAc) units). Larger glycans cannot be directly incorporated due to the formation of side products[25]. As a variant of this method, synthetic linkers can facilitate the attachment of pre-assembled glycans onto peptides, but the linking moieties can be immunogenic[26]. The GlcNAc entities can be extended in vitro by enzymatic transglycosylation. This method relies on modified endo-β-N-acetylglucosaminidase enzymes that add N-glycans to a GlcNAc-asparagine within acceptor sequences[23]. However, the technique rarely reaches above 50% sequon occupancy and requires chemical synthesis to generate sufficient oxazoline donor substrates[6,27,28]. This is generally not feasible for ER-type glycans such as oligomannose and glucosylated ones. In an alternative in vitro approach, we and others have used recombinant enzymes of the N-glycosylation pathway to assemble glycans onto chemically synthesized lipid-linked oligosaccharide (LLO) analogs, and used these moieties to glycosylate model peptides for the study of ER glycosyltransferases[29–36]. Unfortunately, the approach is not scalable and requires synthetic, dolichol-resembling LLO analogs carrying a GlcNAc₂ moiety, which are expensive and challenging to produce[37,38]. As previous studies have pointed out, the lack of enzymatic strategies to afford these LLO precursors represents a major bottleneck for democratizing the use of well-defined eukaryotic N-glycans and N-glycoconjugates[33,35].

To address these challenges, we have developed a modular, fully enzymatic in vitro pipeline that produces peptides and polypeptides carrying homogeneous eukaryotic N-glycans. Our platform, referred to as GLYCO-BUILD, leverages purified recombinant enzymes from bacterial, archaeal, and eukaryotic sources to assemble glycans ranging from GlcNAc₂ to GlcNAc₂Man₉Glc₃. Our pipeline also affords both ER-Man₃ and Golgi-Man₅, the precursors of complex and hybrid N-glycans, respectively. A key component of our pipeline is that the involved enzymes are able to use the commercially sourced lipid carrier phytol instead of the native dolichol. Accomplishing this simplification required extensive screening of expression constructs and enzyme

homologs as well as reaction conditions. The assembled glycans can be transferred en bloc to synthetic peptides or recombinant proteins carrying canonical sequons by using single-subunit oligosaccharyltransferases that can be heterologously expressed with high yields. We applied our enzymatic pipeline to generate a set of diverse products: (i) mannosylated and glucosylated N-glycans that are inaccessible using conventional methodologies at such scale and level of purity; (ii) glycopeptides and proteins modified at one or multiple sequons; (iii) oligomannose-containing peptides mimicking surface epitopes of human viruses. Our work opens avenues for studying cellular N-glycosylation and generating tool compounds for diagnostic and therapeutic investigations[2,39,40].

## Results

### Identification of enzymes processing phytol as a lipid carrier

In eukaryotes, the sequential activity of ALG (Asparagine-Linked Glycosylation) enzymes in the ER membrane mediates the synthesis of the dolichylpyrophosphate-linked GlcNAc₂Man₉Glc₃ glycan that is transferred to acceptor proteins (Fig. 1a)[3]. The regio- and stereoselectivity displayed by ALG enzymes facilitates the accurate formation of the required glycosidic linkages, making these enzymes an attractive option for producing homogeneous N-glycans in vitro. However, some of these enzymes exhibit a strict requirement for dolichol as the lipid carrier[41,42]. Dolichol is an isoprenoid alcohol that is challenging to synthesize and only present at low concentrations in natural sources (Fig. 1b). We therefore investigated whether phytol (abbreviated as Phy), an inexpensive, commercially available 20-carbon alcohol, which displays a different stereochemistry, could replace dolichol in all ALG- and OST-catalyzed reactions (Fig. 1b). We found that not all tested enzymes processed phytol-derived LLOs equally well and therefore performed homolog screening to identify a suitable set of enzymes that allowed us to recapitulate the ER-based glycan biosynthesis pathway using phytol instead of dolichol (Fig. 1a). All proteins used in the pipeline described below displayed good stability and could be frozen, allowing long-term storage for future use. The LLO assembly reactions could be carried out individually or in suitable combinations (referred to as reaction pots) to yield the most interesting glycans from a biotechnological or basic research standpoint (Fig. 1a, c, d). We assessed the correct assembly of the generated oligosaccharides after transferring them to fluorescently labeled peptides and by analyzing them by tricine-SDS-PAGE or LC-MS/MS. Glycan transfer was achieved using purified yeast oligosaccharyltransferase (OST) complex[34] or single-subunit OSTs from Trypanosoma brucei, as reported below and in the "Methods" section.

### Pot 1: enzymatic synthesis of phytyl-PP-GlcNAc₂

The conversion of phytol to phytyl-PP-GlcNAc₂ represented the largest hurdle, as no in vitro enzymatic process is presently available. Pot 1 comprises three reactions (Fig. 1a). First, the undecaprenol kinase UdpK from Streptococcus mutans (SmUdpK) phosphorylates phytol to yield Phy-P. While this enzyme was previously shown to phosphorylate phytol at a similar rate as its native substrate undecaprenol, its use for eukaryotic N-glycan synthesis had not been explored[43]. For the second and third steps, we employed enzymes from the thermophilic archaeon Sulfolobus acidocaldarius, which uses a dolichylpyrophosphate (Dol-PP) carrier similar to that found in eukaryotes[44]. SaAglH is a functional homolog of the human phosphoglycosyltransferase DPAGT1 (ALG7 in yeast), while SaAgl24 is a functional homolog of the eukaryotic ALG13/ALG14 complex[45,46]. The three enzymes SmUdpK, SaAglH, and SaAgl24 were expressed in Escherichia coli and purified in detergent solution (Supplementary Fig. 1). While SmUdpK and SaAglH are integral membrane proteins, SaAgl24 is a membrane-associated protein. We established a one-pot reaction containing the three enzymes, dodecyl maltoside (DDM),

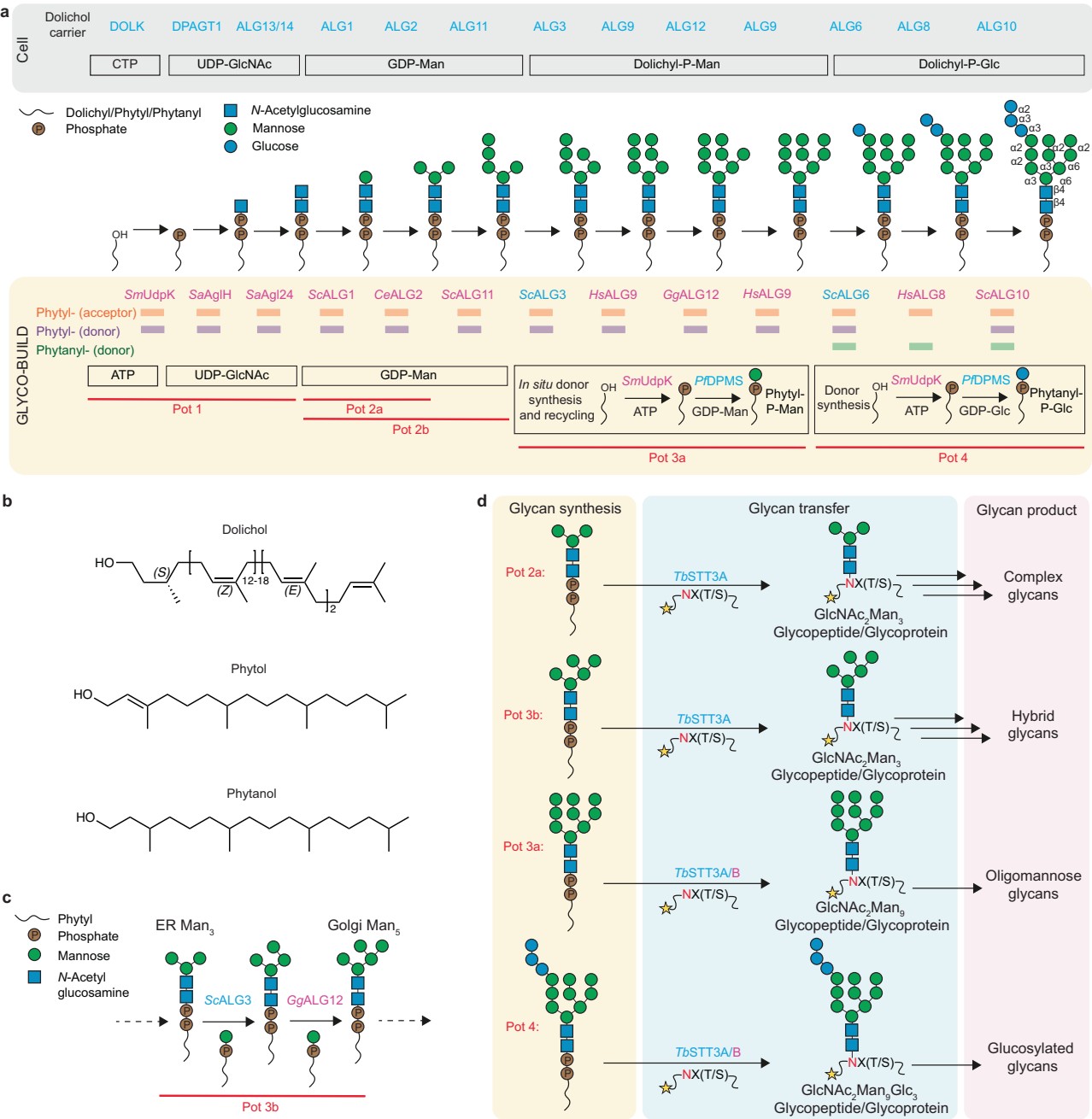

**Fig. 1 | Schematic of GLYCO-BUILD pipeline. a** Comparison of ER-based biosynthesis of dolichylpyrophosphate-linked oligosaccharides and the GLYCO-BUILD pipeline that uses phytol as a lipid carrier. Eukaryotic enzyme names and the required donor substrates are displayed on gray background. Note that DOLK was formerly termed SEC59. Enzymes and donor substrates for the GLYCO-BUILD pipeline are shown on a yellow background. Rectangular boxes indicate that the enzyme homolog reported above is able to process phytyl-linked acceptor substrate (orange) and phytyl- or phytanyl-linked donor substrate (purple and green, respectively). **b** Structures of lipid carriers. Dolichol is used in the *N*-glycosylation pathway of eukaryotes and archaea. Phytol is a diterpene alcohol containing 20 carbons, with the first isoprene unit unsaturated. Phytanol is a fully saturated 20-carbon-long diterpene alcohol. **c** GLYCO-BUILD can be used to synthesize hybrid glycans in vitro, circumventing the in vivo pathway of oligomannose build-up and trimming. **d** GLYCO-BUILD encompasses two steps: glycan synthesis (yellow-shaded) and glycan transfer (blue-shaded). Transfer of the pre-assembled glycans onto glycopeptides and glycoproteins is accomplished using the single-subunit eukaryotic oligosaccharyltransferases *Tb*STT3A and *Tb*STT3B. The yellow star in (**d**) represents an N-terminal fluorophore that facilitates glycoconjugate detection. The asparagine residue that is glycosylated is indicated in red. Glycan products belong to four different categories of interest (pink-shaded). In (**a, c, d**), names in magenta correspond to recombinant enzymes distinct from those reported in the literature (cyan). Glycan structures are depicted using the SNFG (Symbol Nomenclature for Glycans) system.

phytol, ATP, and UDP-GlcNAc, which yielded homogeneous Phy-PP-GlcNAc$_2$ (Pot 1, Figs. 1a, 2a, and 3a, b; Supplementary Table 1). Previous studies demonstrated the selectivity of human ALG13/14 towards the lipidic portion of its substrate, preventing enzymatic synthesis of phytanyl-PP-GlcNAc$_2$[47]. Here, we circumvent this issue

and synthesize a polyisoprenoid-pyrophosphate-linked disaccharide using an entirely enzymatic method and an easily sourced lipid carrier. The reaction yield for Pot 1 and all subsequent steps reached >95% product formation under the conditions reported in the "Methods" section, unless specified otherwise.

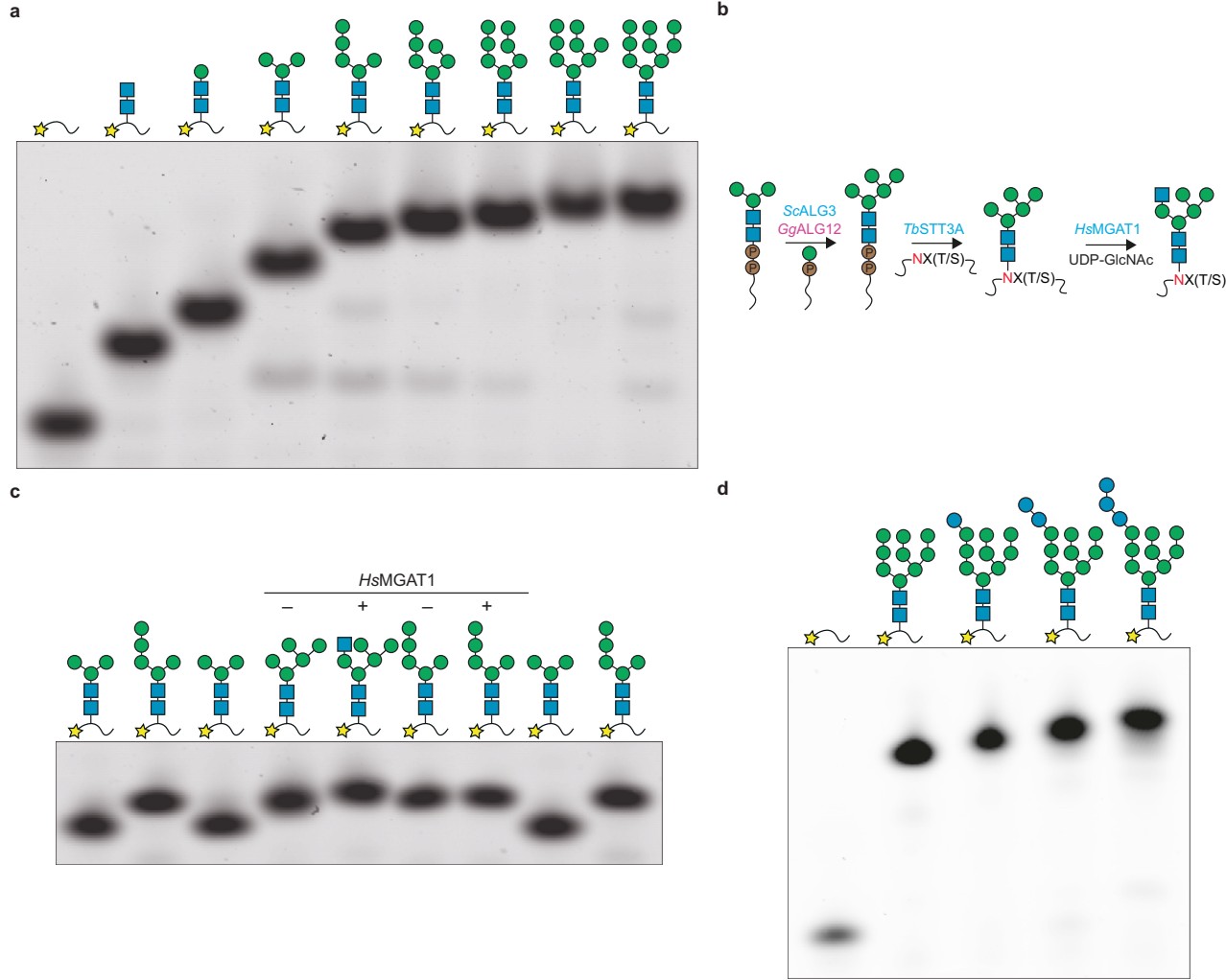

**Fig. 2 | Tricine gel electrophoresis analysis of glycans synthesized and transferred using GLYCO-BUILD.** Following enzymatic LLO synthesis, glycans were transferred to 5-carboxyfluorescein-GSDANYTYTQ using *Tb*STT3A.
**a** Glycopeptides resulting from the synthesis and transfer of oligomannose *N*-glycans. **b** Schematic of Golgi-Man5 synthesis and elongation. **c** Synthesis and analysis of Golgi-Man5 containing glycopeptides. Golgi-Man5 can be extended to GlcNAc2Man5GlcNAc (a hybrid glycan), whereas ER-Man5 cannot. **d** Glycopeptides resulting from the elongation of the A branch of a Man9-containing *N*-glycan with up to three glucose units. Band shift upwards corresponds to the attachment of new sugar units. The glycoforms loaded are indicated above the lanes using the symbol nomenclature for glycans (SNFG), the peptide is depicted by a black line, and the fluorophore is depicted with a yellow star. In (**b**), the asparagine residue that is modified with glycans is indicated in red. Source data are provided as a Source Data file.

## Pots 2a and 2b: synthesis of phytyl-PP-GlcNAc2Man3 or phytyl-PP-GlcNAc2Man5

For the mannosylation of Phy-PP-GlcNAc2, we selected eukaryotic ALG1, ALG2 and ALG11 enzymes (Fig. 1a). We used truncated *Saccharomyces cerevisiae* ALG1 and ALG11 as reported previously[30], but devised purification protocols that yielded higher protein purity and higher activity. Upon screening several ALG2 homologs, we selected *Caenorhabditis elegans* ALG2 because it displayed the highest protein expression and stability. The three enzymes were expressed in *E. coli* and purified by affinity chromatography in detergent solution, which was essential because all three proteins are membrane-associated and contain hydrophobic surfaces that anchor them to the membrane surface (Supplementary Fig. 1). Using GDP-Man as a mannose donor, we generated: (i) Phy-PP-GlcNAc2Man by adding only *Sc*ALG1 to the reaction mixture; (ii) Phy-PP-GlcNAc2Man3 when both *Sc*ALG1 and *Ce*ALG2 were added; (iii) Phy-PP-GlcNAc2Man5 (below abbreviated as ER-Man5 LLO) when *Sc*ALG1, *Ce*ALG2, and *Sc*ALG11 were added to the reaction mixture (Pot 2a, Figs. 1a, 2a, and 3a, b; Supplementary Table 1). Reaction yields exceeded 95% in all three cases.

## Pot 3a: synthesis of oligomannose *N*-glycans

The ER luminal mannosyltransferases ALG3, ALG9 and ALG12 are integral membrane proteins that elongate Dol-PP-GlcNAc2Man5 to Dol-PP-GlcNAc2Man9 in the lumen of the ER in four steps[48]. Previously, *S. cerevisiae* homologs of these enzymes were used for in vitro extension of synthetic dolichol analogs[32,34]. While *Sc*ALG3 yielded satisfactory amounts of high-quality protein, we found that *Homo sapiens* ALG9 and *Gallus gallus* ALG12 yielded higher expression and improved monodispersity (Supplementary Fig. 1). ALG3, ALG9, and ALG12 use dolichylphosphomannose (Dol-P-Man) as a donor substrate, which is not commercially available[32,34]. We therefore established the enzymatic in vitro generation of the phytol analog phytylphosphomannose (Phy-P-Man). This was achieved by mixing phytol, ATP, DDM, and GDP-Man in a coupled reaction containing the kinase *Sm*UdpK and *Pf*DPMS, a single-subunit homolog of the Dol-P-Man synthase from *Pyrococcus furiosus* (Supplementary Fig. 1)[49]. To achieve elongation to Phy-PP-GlcNAc2Man9, we mixed purified *Sc*ALG3, *Hs*ALG9 and *Gg*ALG12 with the acceptor substrate Phy-PP-GlcNAc2-Man5 and an excess of enzymatically generated donor substrate Phy-P-Man in detergent solution (Pot 3a, Figs. 1a, 2a, and 3a, b; Supplementary Table 1). We could

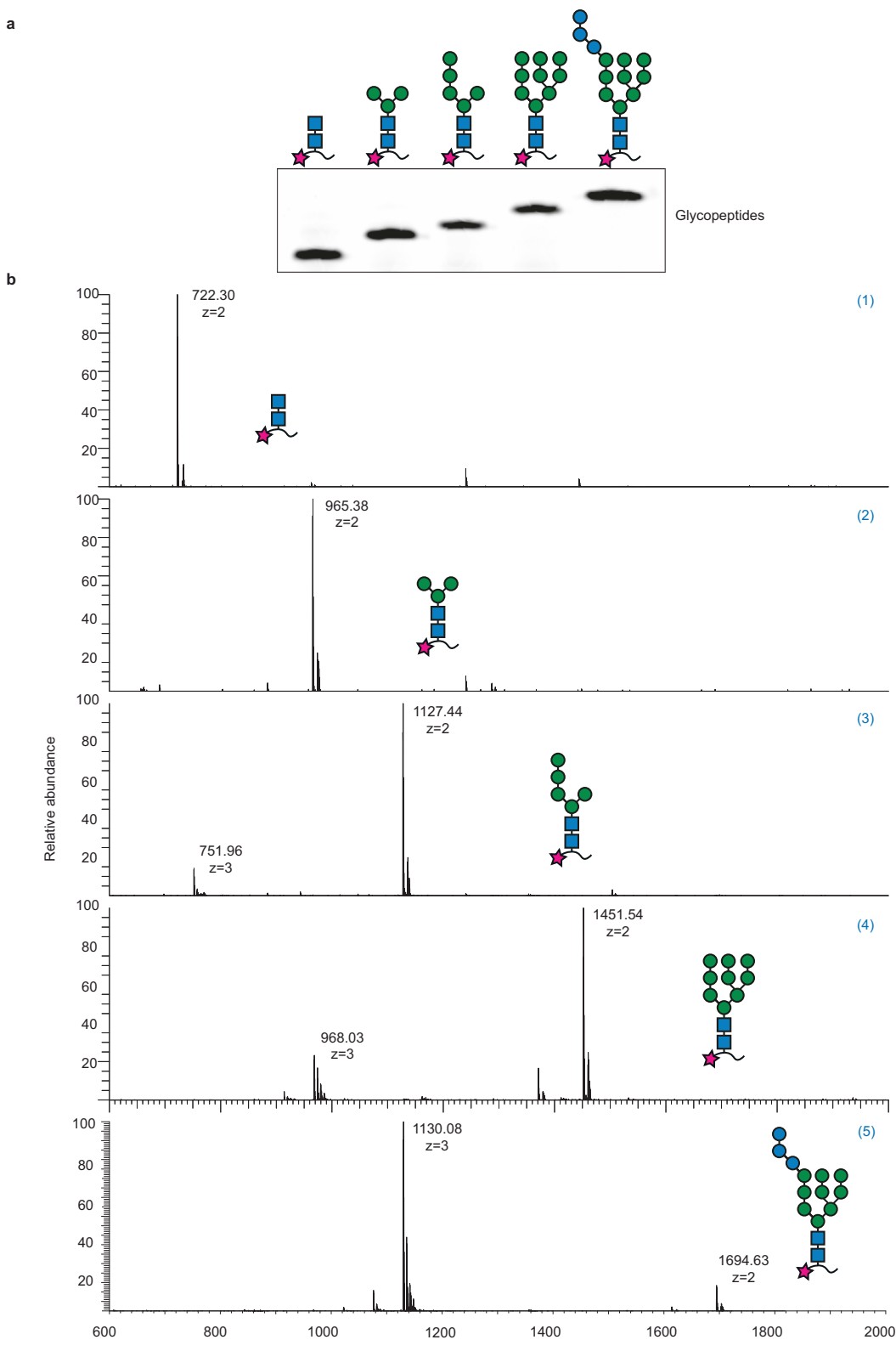

**Fig. 3 | Analysis of enzymatically synthesized glycopeptides. a** Tricine-SDS-PAGE analysis of fluorescently labeled peptide TAMRA-YANATS-NH$_2$ glycosylated in vitro using purified yeast OST and phytyl-PP-linked oligosaccharides. The glycoforms loaded are indicated above the lanes using the symbol nomenclature for glycans (SNFG), the peptide is depicted by a black line, and the fluorophore is depicted with a red star. **b** LC-MS analysis of the glycopeptides shown in (**a**). (1) HRMS (m/z): [M + 2H]$^{2+}$ calcd. for C$_{67}$H$_{86}$N$_{12}$O$_{24}$, 722.30; found, 722.30. (2) HRMS (m/z):

[M + 2H]$^{2+}$ calcd. for C$_{85}$H$_{117}$N$_{12}$O$_{39}$, 965.38; found, 965.38. (3) HRMS (m/z): [M + 2H]$^{2+}$ calcd. for C$_{97}$H$_{136}$N$_{12}$O$_{49}$, 1127.43; found, 1127.43. [M + 3H]$^{3+}$ calcd. for C$_{97}$H$_{136}$N$_{12}$O$_{49}$, 751.96; found, 751.96. (4) HRMS (m/z): [M + 2H]$^{2+}$ calcd. for C$_{121}$H$_{176}$N$_{12}$O$_{69}$, 1451.54; found, 1451.54. [M + 3H]$^{3+}$ calcd. for C$_{121}$H$_{176}$N$_{12}$O$_{69}$, 968.03; found, 968.03. (5) HRMS (m/z): [M + 2H]$^{2+}$ calcd. for C$_{139}$H$_{209}$N$_{12}$O$_{84}$, 1694.62; found, 1694.63. [M + 3H]$^{3+}$ calcd. for C$_{139}$H$_{209}$N$_{12}$O$_{84}$, 1130.08; found, 1130.08. Source data are provided as a Source Data file.

synthesize the oligomannose intermediates Phy-PP-GlcNAc$_2$Man$_6$, Phy-PP-GlcNAc$_2$Man$_7$, and Phy-PP-GlcNAc$_2$Man$_8$, depending on the ALG enzymes included in the reaction mixture (Fig. 2a).

We further optimized the procedure by establishing a continuous Phy-P-Man regeneration system that ran concurrently to the oligomannose $N$-glycan synthesis. We performed Phy-PP-GlcNAc$_2$Man$_5$ elongation reactions using sub-stoichiometric amounts of phytol in the presence of $Sm$UdpK, ATP, $Pf$DPMS and an excess of GDP-Man. We observed complete conversion of Phy-PP-GlcNAc$_2$Man$_5$ to Phy-PP-GlcNAc$_2$Man$_9$, demonstrating that in situ synthesis and regeneration of Phy-P-Man was successful (Figs. 1a, 2a, and 3a, b, Supplementary Table 1). All the elongation reactions tested in Pot 3a achieved >95% conversion yields.

## Pot 3b: synthesis of hybrid $N$-glycan precursors

In the cell, trimming of the Man$_9$ glycan by the action of ER and Golgi mannosidases produces Golgi-Man$_5$, the initial substrate for the assembly of hybrid and complex oligosaccharides[3]. Our pipeline allowed us to circumvent the native pathway of glycan build-up and trimming and to synthesize homogeneous Golgi-Man$_5$ glycan in vitro (Supplementary Fig. 2). Incubation of Phy-PP-GlcNAc$_2$Man$_3$ with $Sc$ALG3, $Gg$ALG12 and excess Phy-P-Man resulted in a glycopeptide that migrated similarly to an ER-Man$_5$ glycopeptide in tricine gel electrophoresis (Pot 3b, Figs. 1c and 2b, c) and featured the correct mass (see below). We next verified whether the presumed Golgi-Man$_5$ glycan could serve as an acceptor substrate for MGAT1-catalyzed GlcNAc modification[3]. Upon incubation with purified $Hs$MGAT1 and excess UDP-GlcNAc, we observed a band shift in gel electrophoresis, suggesting successful transfer of a GlcNAc unit (Fig. 2c). Under the same conditions, an ER-Man$_5$ glycopeptide yielded no change in migration, indicating the latter cannot serve as an acceptor substrate for MGAT1 (Fig. 2c). We further verified the linkages of the Golgi-Man$_5$ glycopeptide by performing a glycan digestion using an α1,2/3 mannosidase. We reasoned that the mannosidase reaction would produce an unbranched GlcNAc$_2$Man$_3$ glycopeptide from Golgi-Man$_5$ but a GlcNAc$_2$Man$_2$ glycopeptide from ER-Man$_5$ (Supplementary Fig. 3a). As expected, the digested ER-Man$_5$ glycopeptide appears to run between the GlcNAc$_2$Man and GlcNAc$_2$Man$_3$ standards, while the digested Golgi-Man$_5$ glycopeptide runs higher (Supplementary Fig. 3b). Finally, we performed liquid chromatography-tandem mass spectrometry (LC-MS/MS) analysis on the Golgi-Man$_5$, Golgi-Man$_5$GlcNAc, and α1,2/3 mannosidase digested Golgi-Man$_5$ glycopeptides (Supplementary Fig. 3c). Masses for the Golgi-Man$_5$ and Golgi-Man$_5$GlcNAc glycopeptides matched the expected ones, while masses for the digested Golgi-Man$_5$ glycopeptide suggested incomplete trimming to either GlcNAc$_2$Man$_4$ or GlcNAc$_2$Man$_3$ (Supplementary Fig. 3c). Pot 3b afforded homogeneous Golgi-Man$_5$ with >95% synthetic efficiency.

## Pot 4: synthesis of glucosylated $N$-glycans

In higher eukaryotes, the glucosyltransferases ALG6, ALG8 and ALG10 each transfer one glucose from dolichylphosphoglucose (Dol-P-Glc) to the A branch of the Man$_9$ glycan[34,48]. To replicate this part of the pathway in vitro, we generated Phy-P-Glc by mixing phytol, ATP, and GDP-Glc with $Sm$UdpK and $Pf$DPMS and combined this reaction with Phy-PP-GlcNAc$_2$Man$_9$ in the presence of purified $Sc$ALG6, $Hs$ALG8, and $Sc$ALG10 in detergent solution (Pot 4, Fig. 1a; Supplementary Fig. 1). To our surprise, we found that $Hs$ALG8 was not able to process the phytol-based donor Phy-P-Glc. Suspecting that the rigidity in the first unsaturated isoprene unit might be responsible, we replaced the donor substrate synthesis using commercial phytanol instead of phytol. This resulted in full conversion of Phy-PP-GlcNAc$_2$Man$_9$ to Phy-PP-GlcNAc$_2$Man$_9$Glc$_3$, demonstrating that phytol was able to replace dolichol as a lipid anchor of the acceptor substrate in all reactions (Pot 4, Figs. 1a, 2d, and 3a, b; Supplementary Table 1). We could generate Phy-PP-GlcNAc$_2$Man$_9$Glc$_1$ and Phy-PP-GlcNAc$_2$Man$_9$Glc$_2$, by using only

$Sc$ALG6 or $Sc$ALG6 and $Hs$ALG8, respectively (Fig. 2d and Supplementary Fig. 4). Synthetic efficiency for Pot 4 reactions exceeded 95%. Although the use of phytanol allowed us to generate glucosylated products, its cost (10–100 times that of phytol) supports the use of phytol as lipid carrier whenever possible.

## $Tb$STT3A and $Tb$STT3B display distinct LLO preferences in vitro

We next evaluated how efficiently the single-subunit OST enzymes $Tb$STT3A and $Tb$STT3B of $T.$ $brucei$ could transfer phytyl-PP-linked glycans to acceptor peptides. These OSTs display distinct donor and acceptor substrate preferences: whereas $Tb$STT3A was reported to have a preference for Dol-PP-GlcNAc$_2$Man$_5$, $Tb$STT3B prefers Dol-PP-GlcNAc$_2$Man$_9$[50]. Furthermore, $Tb$STT3A more efficiently glycosylates sequons flanked by acidic side chains, whereas $Tb$STT3B prefers sequons flanked by neutral or basic residues[51]. In prior work, we have reported a method to over-express active $Tb$STT3A in insect cells[29]. We now established a protocol for the expression and purification of functional $Tb$STT3B, which could be incorporated in our pipeline (Supplementary Fig. 1). We compared the glycosylation activity of $Tb$STT3A and $Tb$STT3B using chemoenzymatically generated Dol20-PP-linked glycans with the enzymatically generated Phy-PP-linked oligosaccharides containing either Man$_5$ or Man$_9$ moieties (Supplementary Fig. 5). Both $Tb$STT3A and $Tb$STT3B catalyzed transfer of Phy-PP-oligosaccharides to acceptor peptides with >95% reaction yields within 1 h. We compared the transfer rates of $Tb$STT3A and $Tb$STT3B using Phy-PP-GlcNAc$_2$Man$_3$, Phy-PP-GlcNAc$_2$Man$_5$ and Phy-PP-GlcNAc$_2$Man$_9$ (Fig. 4). We found that $Tb$STT3A could transfer all three glycans, and the reactions reached completion within 1 h. The transfer rate for GlcNAc$_2$Man$_5$ was higher than that of GlcNAc$_2$Man$_3$ and GlcNAc$_2$Man$_9$, in agreement with the donor substrate preference displayed in vivo[50]. $Tb$STT3B-mediated transfer yielded >95% completion with the donor substrate GlcNAc$_2$Man$_9$, whereas only 15% glycosylation was observed for GlcNAc$_2$Man$_5$, and no transfer was detected for GlcNAc$_2$Man$_3$ (Fig. 4).

## Glycosylation of polypeptides containing multiple glycosylation sequons

We next explored whether we could glycosylate up to three closely spaced glycosylation sequons fused to the terminus of a folded protein. We designed and generated fluorescently labeled fusion constructs of $E.$ $coli$ thioredoxin (Trx) and three C-terminal glycosylation sequons optimized for $Tb$STT3A processing, which we termed Fluo-Trx-(A-sequon)$_3$ (Fig. 5a). The Cys-Gly-Ser linker between Trx and the glycosylation sequons allowed fluorescent labeling with fluorescein-5-maleimide. Titration of donor LLO revealed a ladder in gel electrophoresis, revealing species carrying one, two or three glycans. At excess LLO, we observed complete (>95%) glycosylation of all three acceptor sequons of Fluo-Trx-(A-sequon)$_3$ using either Phy-PP-GlcNAc$_2$Man$_3$, Phy-PP-GlcNAc$_2$Man$_5$ or Phy-PP-GlcNAc$_2$Man$_9$ (Fig. 5b, c, d).

We similarly assessed the ability of $Tb$STT3B to glycosylate a polypeptide at multiple sites. We designed an acceptor peptide construct based on the one used for $Tb$STT3A, but with a small aliphatic residue at the −2 position. This sequon was termed B-sequon (Supplementary Fig. 6a). Under the conditions tested here, $Tb$STT3B was less efficient than $Tb$STT3A in attaching multiple glycans to these closely spaced sequons. Incubation of Fluo-Trx-(B-sequon)$_3$ with $Tb$STT3B and Phy-PP-GlcNAc$_2$Man$_9$ yielded 53% triple-glycosylated and 47% double-glycosylated product as determined by gel densitometry (Supplementary Fig. 6b). In agreement with what we observed in reactions using fluorescently labeled peptides (Fig. 4), $Tb$STT3B could transfer neither GlcNAc$_2$Man$_3$ nor GlcNAc$_2$Man$_5$ efficiently.

## Preparation of viral glycopeptides bearing oligomannose glycans

Short viral glycosylated antigens (glycopeptides) that are able to induce relevant B and T cell responses might find applications in

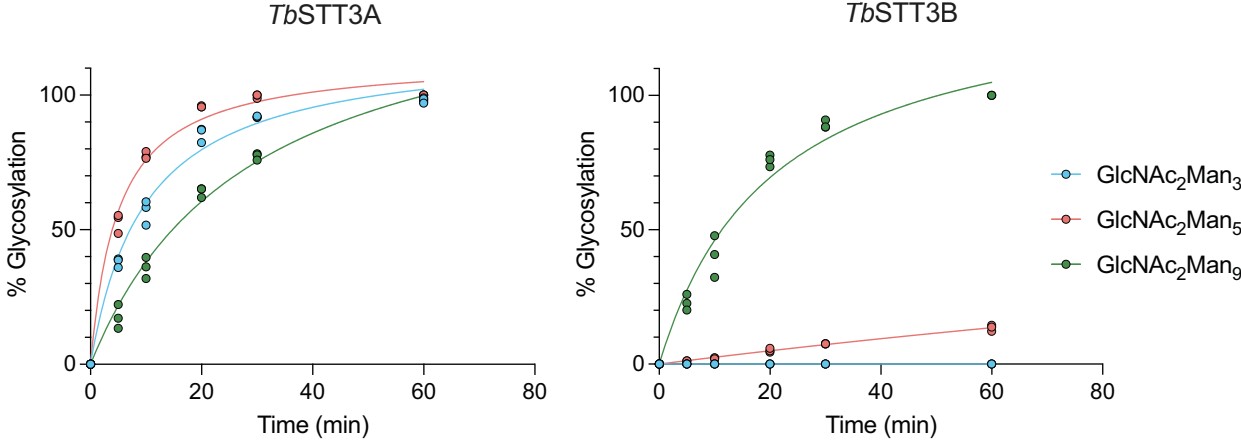

**Fig. 4 | Kinetic studies of glycan transfer using *Tb*STT3A, *Tb*STT3B and phytyl-PP-linked oligomannose structures.** Time course of in vitro glycosylation reactions using 100 nM purified enzyme (*Tb*STT3A or *Tb*STT3B), 15 µM LLO (phytyl-PP-Man$_{3/5/9}$) and 10 µM fluorescently labeled peptide (5-carboxyfluorescein-GSDANYTYTQ for *Tb*STT3A or 5-carboxyfluorescein-GSLANYTK for *Tb*STT3B). Data (*n* = 3 per condition) are presented as individual data points. Source data are provided as a Source Data file.

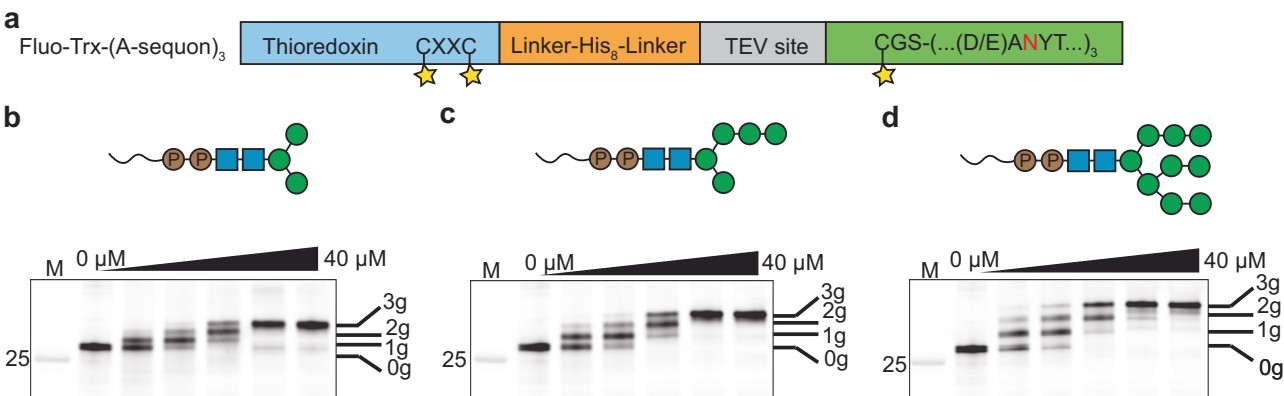

**Fig. 5 | SDS-PAGE analysis of synthesized glycoproteins containing three glycosylation sequons. a** Construct design of a thioredoxin with three C-terminal A-sequons. The glycosylation sequons are shown with the acceptor asparagine colored in red. Cysteine residues are indicated bound to fluorescein, represented by a yellow star. *Tb*STT3A catalyzed glycoprotein synthesis with (**b**) Phy-PP-GlcNAc$_2$Man$_3$, (**c**) Phy-PP-GlcNAc$_2$Man$_5$ or (**d**) Phy-PP-GlcNAc$_2$Man$_9$ glycans. **b**–**d** Fluorescently labeled Trx-(A-sequon)$_3$ was incubated with increasing amounts of LLO (0 µM, 5 µM, 10 µM, 20 µM, 30 µM, 40 µM) and *Tb*STT3A overnight before the products were separated on an SDS-PAGE gel. Bands are labeled with 0 g, 1 g, 2 g, or 3 g to indicate 0, 1, 2, or 3 attached glycans, respectively. Glycoforms used for glycosylation are depicted using the symbol nomenclature for glycans (SNFG). The molecular weight of the fluorescent protein standard is indicated in kDa. Source data are provided as a Source Data file.

diagnostics or immunization studies[14–16]. We therefore tested whether our pipeline facilitated the generation of viral glycopeptide epitopes known to carry oligomannose glycans. We first synthesized a glycopeptide based on the NS1 protein of the Dengue virus because this protein was shown to elicit an IgG response in humans, making it a potential candidate for vaccination studies[9,52]. Within the *Flaviviridae* family, NS1 has a conserved glycosylation sequon at Asn207, which bears an oligomannose glycan and was identified as an area of B- and T-cell epitopes (Fig. 6a, b)[9,10,52–55]. Using Phy-PP-GlcNAc$_2$Man$_9$ and *Tb*STT3A, we converted a synthetic 15-AA peptide comprising the Asn207 sequon to the corresponding glycopeptide with a yield of >95% (Fig. 6b, c).

We next aimed to generate Man$_5$ and Man$_9$ glycoforms for the antigenic site 412 (AS412) that is part of the E2 envelope protein of the Hepatitis C virus (HCV) (Fig. 6d)[56,57]. This viral epitope contains an *N*-glycosylation sequon, which is the target of broadly neutralizing human antibodies (bnAbs) and has been identified as a promising vaccine target[58–60]. We designed two 15-AA peptides, named E2-N417 and E2-N415, which represented the sequence encompassing amino acids 412-423 of AS412. The synthetic peptides contained N-terminal 5-carboxyfluorescein moieties to facilitate detection and a Gly-Ser-Arg sequence to enhance peptide solubility (Fig. 6e). E2-N417 and E2-N415 display two different glycosylation sequons naturally present in HCV variants, which arise through "glycan shifting" and mediate HCV neutralization escape and resistance to bnAbs[11]. By titrating increasing amounts of phytyl-linked LLOs and using either *Tb*STT3A or *Tb*STT3B according to the sequon preference of the enzymes, we achieved homogeneous glycosylation of both E2-N417 and E2-N415 with GlcNAc$_2$Man$_5$ and GlcNAc$_2$Man$_9$ (Fig. 6f, g). In the case of E2-415, glycosylation reached >95% for both glycans; for E2-417, the peptide was completely modified with GlcNAc$_2$Man$_5$ but to a lesser extent (22.5%) with GlcNAc$_2$Man$_9$.

We next tackled the generation of glycopeptides of the HIV envelope protein gp120, which contains several epitopes modified with oligomannose glycans, including the V3 epitope[12,13,61]. Engineered V3 glycopeptides have been shown to bind broadly neutralizing antibodies isolated from infected patients, underscoring their potential for serum screening or vaccination studies (Fig. 6h)[62]. We designed a 35-

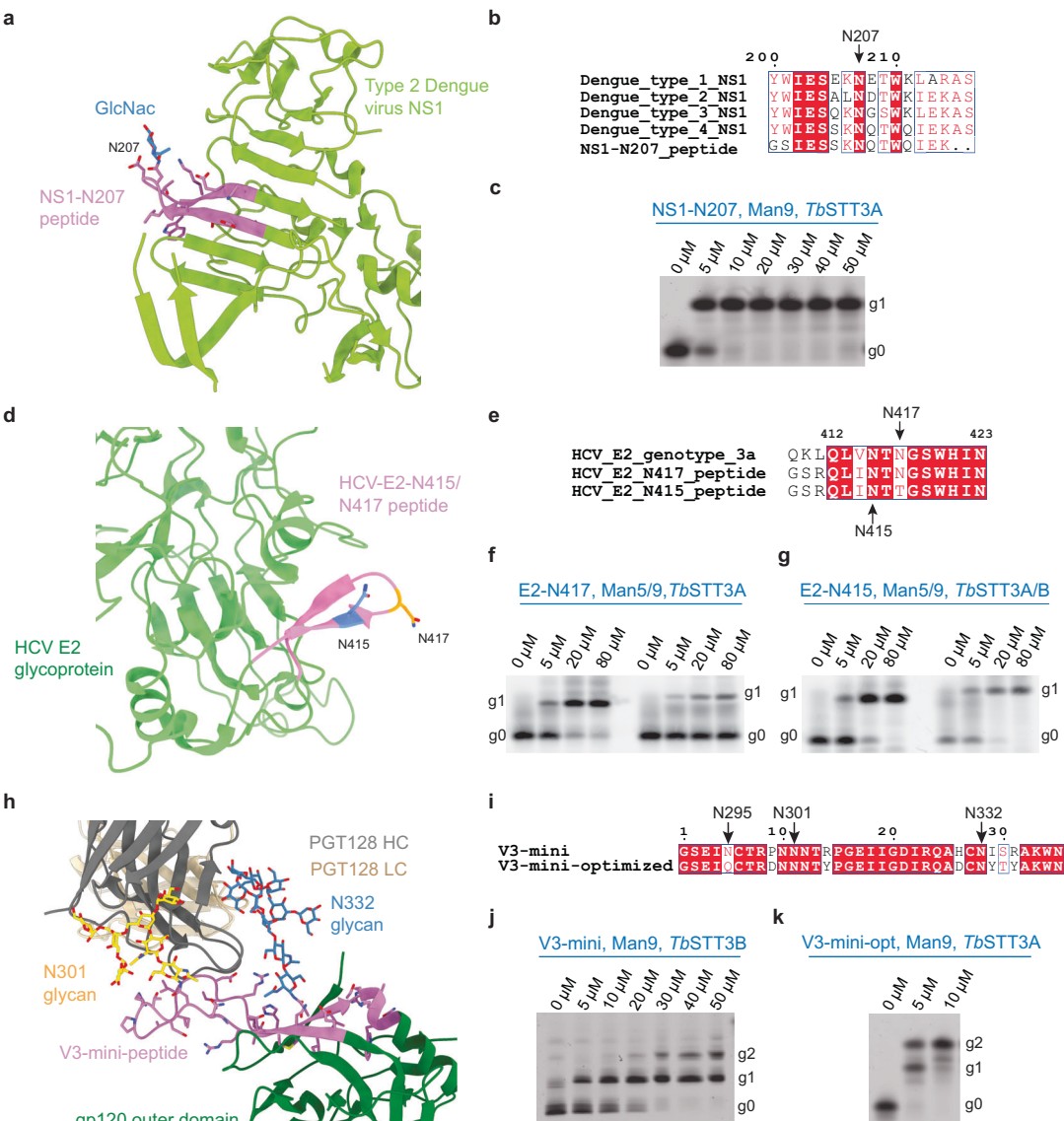

**Fig. 6 | Glycosylation of viral peptides from Dengue, Hepatitis C, and HIV viruses. a** Structure of the NS1 protein from type 2 Dengue virus (PDB ID 4o6b), with the relevant surface epitope containing the N-glycosylation site shown in purple and labeled. **b** Amino acid sequence alignment of NS1 sequences from different Dengue virus types and the synthetic peptide NS1-N207 used in this study. **c** Glycosylation of NS1-N207 peptide with Phy-GlcNAc$_2$Man$_9$ using *Tb*STT3A. LLO concentrations are indicated above the lanes. **d** Structure of the E2 envelope protein monomer of Hepatitis C virus (HCV) (PDB ID 8rk0), with the relevant surface epitope containing the N-glycosylation site shown in purple and labeled. **e** Amino acid sequence alignment of immunogenic HCV E2 epitope (residues 412-423) with the peptides HCV-E2-N417 and HCV-E2-415 used in this study, reflecting sequences before and after resistance-promoting N417T mutation. Glycosylation of (**f**) HCV-E2-N417 and (**g**) HCV-E2-415 peptides with Phy-GlcNAc$_2$Man$_5$ and Phy-GlcNAc$_2$Man$_9$ using either *Tb*STT3A or *Tb*STT3B. **h** Structure of the complex of PGT128 Fab and an engineered gp120 outer domain of HIV (PDB ID: 3tyg), with the relevant surface epitope containing the N-glycosylation site shown in purple and labeled. **i** Amino acid sequence alignment of the V3-mini and V3-mini-optimized peptides used here. Glycosylation of (**j**) V3-mini and (**k**) V3-mini-optimized peptides with Phy-GlcNAc$_2$Man$_9$ using *Tb*STT3B or *Tb*STT3A as indicated. In all panels showing gels, g0, g1, and g2 shown on the right refer to unglycosylated peptide or peptide featuring single or double glycosylation. Source data are provided as a Source Data file.

residue peptide with three glycosylation sequons, termed V3-mini (Fig. 6i)[62]. Using *Tb*STT3B and Phy-PP-GlcNAc$_2$Man$_9$, approximately 29% of the peptide was doubly glycosylated, with rest carrying only one glycan, as determined by gel densitometry (Fig. 6j). Negligible amounts carrying three glycans were observed. We hypothesized that quantitative glycosylation was hindered by the non-optimal acceptor sequences. Given that PGT128 and other Fabs have been shown to interact with only two *N*-glycans (at N301 and N332, but not N295)[12,14,61,62], we generated a second peptide, termed V3-mini-optimized, which contained two acceptor sequons for glycosylation at N301 and N332 and engineered for efficient processing by *Tb*STT3A (Fig. 6i). We observed >95% double glycosylation for this peptide

(Fig. 6k). While conversion to fully glycosylated peptide was not achieved in all tested cases, we envisage purification steps would likely yield pure products.

## Discussion

In this study, we employed a fully enzymatic approach to generate homogeneous eukaryotic *N*-glycans and *N*-glycopeptides, addressing some of the limitations of state-of-the-art methodologies. We identified a set of enzymes that accept phytol as a lipid carrier, allowing the eukaryotic ER-based glycan biosynthesis to be fully recapitulated in vitro. The selected homologs were in part chosen based on earlier studies that suggested enzyme promiscuity with respect to the lipid

carrier used[30,31,42]. For others, we resorted to homolog screening and optimization of reaction conditions, because published enzymes did not accept phytol or were poorly expressed. Strict selectivity for the lipid tail was especially evident in the case of ALG8, which was able to accept phytol as LLO carrier, but required phytanylphosphate-linked glucose as donor substrate. This further demonstrates the complexity in searching for enzymes able to process commercial isoprenoid substrates. By using purified enzymes in user-defined combinations, we gained full control of the generated glycan structures and eliminated impediments associated with glycan and glycopeptide heterogeneity, which are particularly problematic when the target glycosylated molecule is isolated from natural sources or produced with crude enzyme preparations[17,35]. Furthermore, facile synthesis of compounds such as phytyl-PP-GlcNAc$_2$, phytyl-P-mannose, and phytyl-P-glucose will not only facilitate structural and functional studies of N-glycosylation, but also of O- and C-mannosylation as well as glycosylphosphatidylinositol biosynthesis. The current purification yields of functional enzymes fully support the needs for academic research labs and democratize access to compounds unavailable otherwise. We anticipate that further optimization and scale-up of our workflow will enable its transition to a higher-scale setting.

A second advance of our study was to establish conditions under which the single-subunit OSTs from T. brucei, TbSTT3A and TbSTT3B, can process multiple sequons within (poly)peptides. In contrast to multimeric OST enzymes of yeast or higher eukaryotes, optimized constructs of TbSTT3A and TbSTT3B can be recombinantly overexpressed in insect cells and purified in large quantities and in an active form[29,34]. The distinct sequon preference displayed by TbSTT3A and TbSTT3B provides versatility in the choice of the peptide or protein that can be glycosylated with one or even multiple glycan moieties. Furthermore, we demonstrated the ability of these single-subunit OSTs to transfer glucosylated N-glycans onto peptides to near completion, something unexpected given their reported specificities in vivo[50,51].

We expect that GLYCO-BUILD will have broad applicability. First, it allows the generation of specific and homogeneous oligomannose structures. Glycopeptides have shown promising results in eliciting immunological response in animal models of HIV and HCV infections[14,15,63]. We envision that the generation of viral glycopeptides with defined and homogeneous oligomannose glycans could benefit both serum testing and immunological investigations, as such glycoconjugates might recapitulate aspects of the humoral immune response elicited by fully folded viral surface proteins[16]. Second, the GlcNAc$_2$Man$_3$ and Golgi-Man$_5$ glycans generated using our pipeline can serve as starting points for generating complex and hybrid glycopeptides by enzymatic methods developed previously[28,64,65]. This might help improve the pharmacokinetics and pharmacodynamics of peptides and proteins used as biologics, for example, by modulating their half-life[28,64]. Finally, we anticipate that GLYCO-BUILD will find applications where a lack of homogeneous eukaryotic glycans is a limiting factor. For example, trans-glycosylation requires large amounts of homogeneous N-glycans that are converted to oxazoline donors[66]. The LLO synthesis part of our enzymatic method could provide the required donors fully enzymatically and supplement the glycans that remain otherwise inaccessible, including those carrying glucose on the A-branch. Homogeneous LLOs could also be used in combination with existing cell-free technologies to produce glycoproteins on demand[22,67] and to study the pharmacological effect elicited by particular glycoforms, both in vitro and in animal models. We also anticipate our enzymatically assembled glycans to be applicable to the production of glycoarrays or as standards for mass spectrometry libraries. In summary, we envision that our versatile technology will accelerate discovery in glycobiology and facilitate biotechnological applications.

## Methods

### Expression and purification of SmUdpK

A synthetic gene coding for UdpK from S. mutans (Uniprot: Q05888) was codon optimized for expression in E. coli (GeneArt, Thermo Fisher Scientific) and cloned into a modified pET vector with an N-terminal T7 peptide expression tag and a C-terminal His$_{10}$ purification tag. For protein expression, transformed E. coli BL21 (DE3) cells were grown in Luria–Bertani media supplemented with 100 μg/mL ampicillin at 37 °C, until an optical density of 1.0 at 600 nm was reached. Protein expression was induced by addition of 1 mM isopropyl β-D-1-thiogalactopyranoside (IPTG), after which the temperature was set to 16 °C for overnight expression. Cells were harvested by centrifugation, flash-frozen in liquid nitrogen and stored at −80 °C until further use. All purification steps were carried out on ice or at 4 °C. For purification, cells were resuspended using a 1:5 weight to volume ratio of UdpK lysis buffer (50 mM Tris/HCl pH 7.4, 300 mM NaCl, 5% glycerol, 25 mM imidazole pH 8.0, 0.05 mg/mL DNaseI, 1 mM PMSF, complete™ EDTA-free Protease Inhibitor Cocktail (Roche)). Resuspended cells were lysed by three cycles of sonication (50% amplitude, 3 s ON, 3 s OFF, 1 min active sonication time, 2 min break between each sonication cycle) carried out on ice using a Branson Digital Sonifier SFX 250. The lysate was solubilized by the addition of DDM (Anatrace) to a final concentration of 0.5% (w/v), stirring for 1.5 h at 4 °C. Insoluble debris was removed by centrifugation at -158,000 g in a Type Ti45 rotor (Beckman–Coulter) for 40 min. The soluble fraction was collected, filtered with a 0.45 μm filter (Millipore Sigma) and loaded on a 5 mL Superflow Ni-NTA (Qiagen) column, pre-equilibrated in UdpK equilibration buffer (50 mM Tris/HCl pH 7.4, 300 mM NaCl, 5% glycerol, 10 mM MgCl$_2$, 0.03 % (w/v) DDM, 25 mM imidazole pH 8.0). After loading, the resin was washed with 10 CV UdpK wash buffer (50 mM Tris/HCl pH 7.4, 300 mM NaCl, 5% glycerol, 10 mM MgCl$_2$, 0.03 % (w/v) DDM, 50 mM imidazole pH 8.0), before elution in the same buffer but containing 250 mM imidazole pH 8.0. Purified SmUdpK was immediately desalted using PD-10 columns (Cytiva), pre-equilibrated in UdpK desalting buffer (50 mM Tris/HCl pH 7.4, 300 mM NaCl, 10 mM MgCl$_2$, 0.03 % DDM), and concentrated using a 50 kDa molecular weight cutoff Amicon centrifugal filter (Merck Millipore). Protein aliquots were flash-frozen in liquid nitrogen and stored at −80 °C until further use. Protein purity and oligomeric state were confirmed by SDS-PAGE and size-exclusion chromatography using a Superdex S200 10/300 GL (GE Healthcare) in UdpK desalting buffer, respectively.

### Expression and purification of SaAglH and SaAgl24

Genes from S. acidocaldarius coding for AglH (Uniprot ID: P39465) and Agl24 (Uniprot: Q4J9C3) were codon optimized for expression in E. coli using GeneArt (Thermo Fisher Scientific) and cloned into a modified pET vector with a C-terminal GSS linker, HRV 3C-cleavage site and His$_8$ purification tag. Both constructs were expressed using E. coli BL21 (DE3) cells grown in ZYP-5052 autoinduction media[68] at 37 °C for 4 h followed by expression overnight at 16 °C. Cells were harvested by centrifugation, flash-frozen in liquid nitrogen and stored at −80 °C until needed. Purification of both constructs was carried out with the same purification scheme at 4 °C. Cells were resuspended in Agl lysis buffer (50 mM HEPES, pH 7.5; 500 mM NaCl) in a 1:5 weight to volume ratio, and phenylmethylsulfonyl fluoride and DNAse I (Roche) were added to final concentrations of 1 mM and 5 μg mL$^{-1}$, respectively. Cells were lysed with one pass in a microfluidizer (M-110 P, Microfluidics) with a 200 μm ceramic cell equilibrated in lysis buffer and run at 15,000 psi. Cellular debris was removed by centrifugation at 15,000 × g for 30 min. Membranes from the resulting supernatant were then pelleted by centrifugation at -158,000 × g in a Type 45Ti rotor (Beckman–Coulter) for 1 h. Membranes were resuspended in lysis buffer, flash-frozen in liquid nitrogen and stored at −80 °C.

Isolated membranes were extracted with 1% (w/v) DDM (Anatrace) for 1 h before insoluble components were pelleted at -125,000 g in a

Type 45Ti rotor (Beckman–Coulter) for 30 min. Imidazole stock solution at pH 7.5 was used in the following steps. The supernatant was supplemented with 20 mM imidazole before loading on a 5 mL Superflow Ni-NTA (Qiagen) column equilibrated in Agl buffer (20 mM HEPES, pH 7.5; 500 mM NaCl; 0.017 % (w/v) DDM) also supplemented with 20 mM imidazole. Resin was washed with 50 mM imidazole in Agl buffer before protein elution in the same buffer but with 250 mM imidazole. Fractions containing the protein of interest were pooled and immediately desalted into Agl desalting buffer (20 mM HEPES, pH 7.5; 150 mM NaCl; 0.017 % (w/v) DDM). Protein was concentrated using a 50 kDa molecular weight cut-off Amicon centrifugal filter (Merck Millipore) and flash frozen in liquid nitrogen.

### Expression and purification of ScALG1, CeALG2, and ScALG11

Expression and purification of ScALG1 residues (Uniprot ID: P16661, residues 33–349), and ScALG11 (Uniprot ID: P53954, residues 46–548) were optimized from previously published work[30]. Truncated genes coding for ScALG1 and ScALG11 proteins were cloned into pQE80 (Qiagen) with an N-terminal His$_{10}$ tag followed by two copies of the Z-domain and a TEV cleavage sequence[30]. Expression of ScALG1 was carried out in E. coli BL21 (DE3) grown in ZYP-5052 autoinduction media[68] at 37 °C for 4 h followed by expression overnight at 18 °C. Expression of ScALG11 was performed in E. coli BL21 (DE3) grown in Terrific Broth (TB) media. Cells were grown at 37 °C until an optical density of 1.3–1.5 at 600 nm was reached. The temperature was then reduced to 16 °C, and protein expression was induced with 0.5 mM IPTG for 16 h. Following cell harvesting, all steps were performed at 4 °C. Cells were resuspended in lysis buffer (50 mM HEPES, pH 7.5; 150 mM NaCl; 5 mM β-mercaptoethanol, 20 ng mL$^{-1}$ DNase I (Roche); 1 mM PMSF; EDTA-free complete protease inhibitor (Roche)) and lysed in a microfluidizer (M-110 P, Microfluidics) at 15,000 psi with a 200 μm ceramic cell equilibrated in lysis buffer. ALG enzymes were solubilized by adding 1% DDM (Anatrace) and 0.1% cholesteryl hemisuccinate (CHS, Anatrace) for 1 h of stirring. Cellular debris was removed by centrifugation at ~125,000 g in a Type 45Ti rotor (Beckman–Coulter) for 30 min. Imidazole pH 8.0 was added to the supernatant to a final concentration of 25 mM before loading onto a 5 mL column of Superflow NTA resin (Qiagen) equilibrated in ALG1 buffer containing 20 mM HEPES, pH 7.5; 150 mM NaCl; 5 mM β-mercaptoethanol; 0.03% (w/v) DDM; 0.006% (w/v) CHS, and 25 mM imidazole pH 8.0. The resin was washed with ALG1 buffer containing with 50 mM imidazole pH 8.0 before the protein was eluted with ALG1 buffer containing 300 mM imidazole. The eluted protein was immediately desalted into buffer ALG1 buffer, concentrated on a 50 kDa molecular weight cut-off Amicon centrifugal filter (Merck Millipore), flash frozen in liquid nitrogen and stored at −80 °C.

The gene from Caenorhabditis elegans coding for ALG2 (Uniprot ID: Q19265) was codon optimized for expression in E. coli using GeneArt (Thermo Fisher Scientific) and cloned into a modified pET-19b vector with a C-terminal His$_{10}$ affinity tag. CeALG2 was expressed using E. coli BL21-Gold (DE3). Cells were grown in modified TB medium supplemented with 1% glucose (w/v) at 37 °C until an OD$_{600}$ of 2.7–3.0 was reached. Protein expression was induced by the addition of 0.5 mM IPTG, after which the temperature was set to 25 °C for 1 h. Cells were harvested by centrifugation, flash-frozen in liquid nitrogen and stored at −80 °C until needed. All the following steps were performed at 4 °C.

Cells were resuspended in ALG2 lysis buffer (50 mM HEPES, pH 7.0; 250 mM NaCl; 3 mM β-mercaptoethanol; 10% glycerol; and 0.5 mM PMSF) using a 1:8 weight-to-volume ratio. Cell lysis was performed using a microfluidizer (M-110 P, Microfluidics) at 15,000 psi with a 200 μm ceramic cell equilibrated in lysis buffer. Membranes and cell debris were pelleted by ultracentrifugation at ~95,000 g in a Type 45Ti rotor (Beckman–Coulter) for 30 min, resuspended in lysis buffer using a 1:5 weight (cell pellet) to volume ratio, flash frozen in liquid nitrogen and stored at −80 °C.

For purification, membranes were solubilized with 1% DDM (Anatrace) and 0.1% CHS (Anatrace) for 1 h at 4 °C while stirring. Insoluble debris was removed by centrifugation at ~95,000 g in a Type 45Ti rotor (Beckman–Coulter) for 30 min. Imidazole pH 8.0 was added to the supernatant to a final concentration of 25 mM before loading onto a 5 mL column of Superflow NTA resin (Qiagen) equilibrated in ALG2 buffer (50 mM HEPES, pH 7.0; 250 mM NaCl; 3 mM β-mercaptoethanol; 25 mM imidazole pH 8.0; 0.02% (w/v) DDM; 0.002% (w/v) CHS). The resin was washed with ALG2 buffer containing 60 mM imidazole, pH 8.0, before the protein was eluted with ALG2 buffer containing 200 mM imidazole, pH 8.0. The eluted protein was immediately desalted into a buffer containing 25 mM HEPES, pH 7.0; 150 mM NaCl; 3 mM β-mercaptoethanol; 0.02% (w/v) DDM and 0.002% (w/v) CHS, concentrated on a 50 kDa molecular weight cut off Amicon centrifugal filter (Merck Millipore), flash-frozen in liquid nitrogen and stored at −80 °C.

### Expression and purification of PfDPMS

The gene from Pyrococcus furiosus coding for DPMS (Uniprot ID: Q8U4M3) was codon optimized for expression in E. coli using GeneArt (Thermo Fisher Scientific) and cloned into a modified pET-19b vector with a His$_{10}$ affinity tag fused to the N-terminus. Expression and purification were performed as previously described[49]. Briefly, PfDPMS was expressed using E. coli BL21-Gold (DE3). Cells were grown in modified TB medium supplemented with 1% glycerol (w/v) at 37 °C to an OD$_{600}$ of 2.7–3.0, before expression was induced by the addition of 1 mM IPTG, after which the temperature was adjusted to 18 °C for 16 h. Cells were harvested by centrifugation, flash-frozen in liquid nitrogen, and stored at −80 °C until needed. All the following steps were performed at 4 °C. Cells were resuspended in lysis buffer (25 mM NaH$_2$PO4/K$_2$HPO$_4$, pH 7.0; 150 mM NaCl and 1 mM PMSF). Cell lysis was performed in a microfluidizer (M-110 P, Microfluidics) at 15,000 psi with a 200 μm ceramic cell equilibrated in lysis buffer. Membranes were pelleted by ultracentrifugation at ~95,000 g in a Type 45Ti rotor (Beckman–Coulter) for 30 min, resuspended in lysis buffer, flash frozen in liquid nitrogen, and stored at −80 °C. Membranes were solubilized with 1% DDM (Anatrace) for 1 h. Insoluble debris was removed by centrifugation at ~95,000 g in a Type 45Ti rotor (Beckman–Coulter) for 30 min. Imidazole pH 8.0 was added to the supernatant to 25 mM before loading onto a 5 mL column of Superflow NTA resin (Qiagen) equilibrated in DPMS buffer (50 mM NaH$_2$PO$_4$/K$_2$HPO$_4$, pH 7.0; 200 mM NaCl; 0.02% (w/v) DDM). The resin was washed with DPMS buffer containing 50 mM imidazole, pH 8.0, before the protein was eluted with DPMS buffer containing 200 mM imidazole, pH 8.0. The eluted protein was immediately desalted into a buffer containing 50 mM HEPES, pH 7.0; 200 mM NaCl; 10 mM EDTA; 10 mM EGTA and 0.02% (w/v) DDM, concentrated using a 50 kDa molecular weight cut-off Amicon centrifugal filter (Merck Millipore), flash-frozen in liquid nitrogen and stored at −80 °C.

### Expression and purification of ScALG3, HsALG9, and GgALG12

Gene sequences coding for S. cerevisiae ALG3 (Uniprot ID: P38179), Homo sapiens ALG9 isoform 1 (Uniprot ID: Q9H6U8) and G. gallus ALG12 (Uniprot ID: F1P077) were codon optimized for expression in human cells using GeneArt (Thermo Fisher Scientific) and cloned into a modified pUC57 vector using restriction-free cloning[69]. ScALG3 was cloned with a C-terminal HRV 3C-cleavage sequence followed by eYFP and 1D4 sequences. HsALG9 and GgALG12 where cloned with an N-terminal FLAG tag followed by the eYFP sequence and an HRV 3C-cleavage sequence. All three proteins were transiently expressed in suspension HEK293 EBNA cells maintained at 37 °C in humidified incubators with supplemental carbon dioxide. Protein expression was induced via transient transfection with linear polyethylenimine. Plasmid DNA was added to linear polyethylenimine in a 1:2 mass ratio prior to adding 1 μg plasmid DNA per million cells. Cells expressing protein

for 2 days were collected by centrifugation, washed with phosphate-buffered saline (137 mM sodium chloride, 2.7 mM potassium chloride, 10 mM dibasic sodium phosphate, 1.8 mM monobasic potassium phosphate, pH 7.4) and flash frozen before being stored at −80 °C until needed. Cells were thawed in ALG3 lysis buffer (150 mM sodium chloride, 50 mM HEPES pH 7.5, 10% (v/v) glycerol) in a 1:5 (w/v) ratio. All subsequent steps were either performed on ice or at 4 °C. Prior to lysis by dounce homogenization, 1 mM phenylmethylsulfonyl fluoride, 20 μg/mL DNAse I (Roche) and a 1:100 (v/v) dilution of Protease inhibitor cocktail (Sigma) were added. Membrane solubilization was performed for 1 h in 1% (w/v) n-dodecyl-β-D-maltopyranoside (DDM), (w/v) 0.2% CHS with stirring. The lysate was then centrifuged at -125,000 g for 30 min in a Type-45Ti (Beckman Coulter) rotor to pellet insolubilized membranes. The supernatant was incubated with either M2 Flag antibody resin (Sigma) for ALG9 and ALG12 or with 1D4 antibody resin (The University of British Columbia) for ALG3 for 1 h with rotation. The flow-through was discarded, and the resin was washed twice with 15 column volumes (CV) wash buffer (150 mM sodium chloride, 20 mM HEPES pH 7.5, 10% (v/v) glycerol, and 0.017 % DDM, 0.0035 % CHS). HRV 3 C protease was added to the column and incubated for 1 h before the protein of interest was eluted from the column with wash buffer. The protein was concentrated using a 50 kDa molecular weight cut-off Amicon centrifugal filter (Merck Millipore). Size exclusion chromatography (SEC) was performed using a Superdex 200 increase 10/300 column (GE Healthcare) equilibrated in SEC buffer (150 mM sodium chloride, 20 mM HEPES pH 7.5, 0.017 % DDM, 0.0035 % CHS) and run at 0.5 mL min⁻¹. Fractions containing the protein of interest were concentrated as before and either directly used or flash frozen in liquid nitrogen and stored at −80 °C.

### Expression and purification of ScALG6, HsALG8, and ScALG10
Codon-optimized S. cerevisiae ALG6 (Uniprot ID: Q12001) carrying an N-terminal FLAG tag followed by eYFP and an HRV 3C protease-cleavage site was expressed and purified as previously described and as reported below[32]. For HsALG8 (Uniprot ID: Q9BVK2) and ScALG10 (Uniprot ID: P50076), construct designs, expression conditions and purification protocols were optimized from prior work[34]. HsALG8 was cloned into a pOET1 vector (Oxford Expression Technologies) encoding a C-terminal HRV 3C protease-cleavage site, followed by eYFP and a 1D4 purification tag. ScALG10 was cloned into a modified pUC57 vector, containing an N-terminal FLAG tag followed by eYFP and an HRV 3C protease-cleavage site. For ScALG6 and HsALG8 expression, baculoviruses were generated using flashBAC DNA (Oxford Expression Technologies) according to the manufacturer's instructions to infect Spodoptera frugiperda cells. Sf9 cells, at a cell density of $2.0 \times 10^6$ cells/mL, were infected with generation V2 of viruses at a 1:200 (v/v) ratio and incubated at 27 °C in a shaking incubator for 72 h. Cells were harvested by centrifugation at $6500 \times g$, washed with phosphate-buffered saline (PBS: 137 mM sodium chloride, 2.7 mM potassium chloride, 10 mM dibasic sodium phosphate, 1.8 mM monobasic potassium phosphate, pH 7.40) and stored at −80 °C until needed. For ScALG10 expression, HEK293 EBNA cells were transiently transfected using a mixture of plasmid DNA and linear polyethyleneimine (1:2 w/w) and incubated at 37 °C in a humidified shaking incubator with 6% carbon dioxide. After 48 h, cells were harvested by centrifugation, washed with phosphate-buffered saline (137 mM sodium chloride, 2.7 mM potassium chloride, 10 mM dibasic sodium phosphate, 1.8 mM monobasic potassium phosphate, pH 7.4) and stored at −80 °C until needed.

For purification of ScALG6, HsALG8, and ScALG10, all steps were either performed on ice or at 4 °C. Cell pellets were thawed and resuspended in lysis buffer (150 mM sodium chloride, 50 mM HEPES pH 7.5, 10% (v/v) glycerol) supplemented with 1 mM phenylmethylsulfonyl fluoride, 100 μg/mL soybean trypsin inhibitor (Sigma), 100 μg/mL DNAse I (Roche) and cOmplete protease inhibitor

cocktail tablets (Roche; one tablet per 10 g of cells). Cells were lysed by Dounce homogenization, and membranes were solubilized by adding DDM and CHS detergents to final concentrations of 1% and 0.2% (w/v), respectively, followed by incubation for 1 h at 4 °C with gentle stirring. The solubilized lysate was spun down at -125,000 g for 30 min in a Type-45Ti rotor (Beckman Coulter) to pellet insoluble membranes and cellular debris. For ScALG6 and ScALG10, the collected supernatant was incubated with M2 Anti-Flag affinity resin (Sigma), while for HsALG8 Anti-1D4 affinity resin (The University of British Columbia) was used. After 1 h of binding at 4 °C, the flow-through was discarded and the resin washed four times with 5 column volumes (CV) of wash buffer (150 mM sodium chloride, 20 mM HEPES pH 7.5, 10% (v/v) glycerol, and 0.017 % DDM, 0.0035 % CHS). To start elution, in-house produced HRV 3C protease was added to the column and incubated for 1 h, after which the protein of interest was collected. Eluted proteins were concentrated using a 50 kDa molecular weight cut-off Amicon centrifugal filter (Merck Millipore) before undergoing SEC on a Superdex 200 increase 10/300 column (GE Healthcare) equilibrated in SEC buffer (150 mM sodium chloride, 20 mM HEPES pH 7.5, 0.017 % DDM, 0.0035 % CHS). Fractions containing the protein of interest were concentrated as described above and either used immediately or flash frozen in liquid nitrogen for storage at −80 °C.

### Expression and purification of TbSTT3A and TbSTT3B
Expression of a synthetic gene coding for an N-terminal His$_{10}$-YFP-3C tag followed by the T. brucei STT3A sequence was performed as previously described[29] and reported below. A synthetic gene encoding T. brucei STT3B was optimized for expression in insect cells and purchased from GeneScript. TbSTT3B gene was cloned into a pOET1 vector with an N-terminal His$_{10}$-YFP-3C tag. Baculovirus production for both constructs was performed using flashBAC DNA (Oxford Expression Technologies) in S. frugiperda (Sf9) cells following the manufacturer's instructions. For expression, Sf21 cells were infected at a cell density of $2.0 \times 10^6$ cells/mL and incubated for 48 h. Cells were collected by centrifugation at $6500 \times g$ and washed with phosphate-buffered saline (137 mM sodium chloride, 2.7 mM potassium chloride, 10 mM dibasic sodium phosphate, 1.8 mM monobasic potassium phosphate, pH 7.4). Cell pellets were frozen in liquid nitrogen and stored at −80 °C until the time of use.

Purification of TbSTT3A and TbSTT3B, was performed as previously described for TbSTT3A[29]. Briefly, cell pellets were thawed and resuspended in lysis buffer (25 mM K$_2$HPO$_4$/NaH$_2$PO$_4$, pH 7.0; 250 mM NaCl; 10% w/v Glycerol) supplemented with cOmplete™, EDTA-free Protease Inhibitor Cocktail (Roche), using a 1:8 weight to volume ratio, followed by lysis on ice with a dounce homogenizer. The cell lysate was then solubilized with 1% (w/v) DDM, 0.2% (w/v) CHS for 2 h at 4 °C while stirring. After high-speed centrifugation (- 95,000 g, Ti45 rotor, 30 min), the supernatant was loaded onto a Ni/NTA super flow affinity column (Qiagen), washed with the same lysis buffer but containing 50 mM imidazole pH 8.0, 0.035% (w/v) DDM, 0.007% (w/v) CHS and eluted with the same buffer but containing 200 mM imidazole pH 8.0. The protein was desalted into 20 mM HEPES pH 7.5; 150 mM NaCl; 5% glycerol (v/v); 0.035% (w/v) DDM; 0.007% (w/v) CHS using a HiPrep 26/10 column (GE Healthcare). The purification tags were kept in both proteins for glycosylation reactions aimed at following glycan elongation. For kinetic studies, TbSTT3A and TbSTT3B were further incubated with HRV 3C protease overnight at 4 °C. Purification tags and HRC 3C protease were removed by a reverse Ni/NTA step, and TbSTT3A and TbSTT3B were further purified by SEC (Superdex S200 10/300 GL, GE Healthcare) in a buffer containing 20 mM HEPES pH 7.5; 150 mM NaCl; 5% glycerol (v/v); 0.035% (w/v) DDM; 0.007% (w/v) CHS.

### Expression and purification of ScOST
Expression and purification of ScOST were performed as previously described[34,70]. Briefly, 6L of yeast culture (MAT α his3Δ1 leu2Δ0 lys2Δ0

ura3Δ0 arg4Δ0 ost6::LEU2MX6 OST4-1D4::kanMX6 YEp352-OST3) were grown at 30 °C in synthetic dropout medium lacking uracil (6.7 g/l yeast nitrogen base w/o amino acids, 20 g/l glucose and 1.92 g/l SD supplement -Ura, 20 g/l glucose and 1.92 g/l SD supplement -Ura) until an optical density of 3.0–4.0 at 600 nm was reached. The cells were harvested by centrifugation at 5500 × $g$, and the cell pellets were flash-frozen in liquid nitrogen and stored at −80 °C until use. Cells were resuspended in ice-cold lysis buffer containing 50 mM HEPES, pH 7.5, 200 mM NaCl, and 1 mM MgCl$_2$, supplemented with protease inhibitors (EDTA-free protease inhibitor cocktail (Roche), 2.6 mg/l aprotinin, 5 mg/l leupeptin, 1 mg/l pepstatin, 2 mM benzamidine HCl) using a 1:1 weight/volume ratio. The cells were lysed in a BioSpec Beadbeater with 1 min/5 min, on/off cycles (4 cycles) in a water-ice bath, and all following steps were performed at 4 °C. Unbroken cells were removed by centrifugation for 10 min at 3000 × $g$. Membranes were collected by ultracentrifugation at 150,000 × $g$ for 1 h, resuspended in membrane solubilization buffer containing 50 mM HEPES, pH 7.5, 500 mM NaCl, 1 mM MgCl$_2$, 1 mM MnCl$_2$, 10% (v/v) glycerol, and subsequently incubated with a mixture of 1% (w/v) DDM and 0.2% (w/v) CHS for 1.5 h. The supernatant was collected after high-speed centrifugation at 150,000 × $g$ for 30 min and incubated with Sepharose-coupled Rho-1D4 antibody (University of British Columbia) for 3 h. The beads were washed with solubilization buffer containing 0.03% (w/v) DDM and 0.006% (w/v) CHS and subsequently with washing buffer containing 50 mM HEPES, pH 7.5, 250 mM NaCl, 1 mM MgCl$_2$, 1 mM MnCl$_2$ 10% (v/v) glycerol, 0.03% (w/v) DDM, and 0.006% (w/v) CHS. OST complex was eluted with 0.5 mg/mL 1D4 peptide (GenScript) in washing buffer, concentrated, and further purified by SEC using a Superose 6 column (GE Life Sciences) with a buffer containing 50 mM HEPES, pH 7.5; 150 mM NaCl, 1 mM MgCl$_2$, 1 mM MnCl$_2$, 5% (v/v) glycerol, 0.03% (w/v) DDM and 0.006% (w/v) CHS.

### Expression, purification, and fluorescent labeling of Trx-(A-sequon)$_3$ and Trx-(B-sequon)$_3$ constructs

Protein constructs to test protein glycosylation were expressed with a N-terminal *E. coli* thioredoxin tag and three glycosylation sequons. The thioredoxin N-terminus was followed by a linker to a His$_8$ tag, another linker to a TEV cleavage sequence and a CGS linker to the peptide sequence containing the three glycosylation sequons. Two different constructs were designed based on the substrate preferences for *Tb*STT3A (GSDANYTYTQSEKSAASEANYTYSAEGRGSESDANYTYTK) and *Tb*STT3B (GSLANYTYTQSEKSDASLVNYTYSSEGRGSESLANYTY-TEK). The DNA sequences were codon optimized for expression in *E. coli* (IDT web app) and were cloned into a modified pET vector using restriction-free cloning[69].

Both constructs were expressed using *E. coli* BL21 (DE3) cells grown in ZYP-5052 autoinduction media[68] at 37 °C for 4 h followed by expression overnight at 16 °C. Cells were harvested by centrifugation, flash frozen in liquid nitrogen and stored at −80 °C until needed. Purification of both constructs was carried out with the same protocol at 4 °C. Cells were resuspended in lysis buffer (50 mM HEPES, pH 7.5; 500 mM NaCl; 10% glycerol; 1 mM TCEP) in a 1:5 weight to volume ratio and phenylmethylsulfonyl fluoride and DNAse I (Roche) were added to final concentrations of 1 mM and 5 μg mL$^{-1}$, respectively. Cells were lysed with one pass in a microfluidizer (M-110 P, Microfluidics) with a 200 μm ceramic cell equilibrated in lysis buffer and run at 15,000 psi. The lysate was then incubated at 60 °C for 30 min before cellular debris and aggerated protein was removed by centrifugation at -125,000 × $g$ in a Type 45Ti rotor (Beckman−Coulter) for 30 min. Imidazole stock at pH 7.5 was used here. The supernatant was supplemented with 20 mM imidazole before loading on a 5 mL Superflow Ni-NTA (Qiagen) column equilibrated in purification buffer (20 mM HEPES, pH 7.5; 150 mM NaCl; 1 mM TCEP) with 20 mM imidazole added. Resin was washed with 50 mM imidazole in purification buffer, before protein elution in purification buffer with 250 mM imidazole.

Fractions containing the protein of interest were pooled and immediately desalted into desalting buffer (20 mM HEPES, pH 7.5; 1 mM TCEP, 0.5 mM EDTA). Protein was concentrated on a 10 kDa molecular weight cut-off Amicon centrifugal filter (Merck Millipore) and flash frozen in liquid nitrogen.

The Trx-(A-sequon)$_3$ and Trx-(B-sequon)$_3$ construct cysteine residues were fluorescently labeled to facilitate detection on SDS-PAGE analysis using fluorescein-5-maleimide (Thermo Fisher). An aliquot of frozen protein was thawed, and a 100-fold molar excess of TCEP was added and incubated at room temperature for -1 h before desalting into reaction buffer (150 mM NaCl, 20 mM HEPES, pH 7.2). A 25-fold molar excess of fluorescein-5-maleimide dissolved in DMSO was added to the protein and incubated overnight at 4 °C in the dark. The reaction mixture was concentrated on a 10 kDa molecular weight cut off Amicon centrifugal filter (Merck Millipore) before SEC on a Superdex 200 increase column 10/300 (GE healthcare) at 0.5 mL min$^{-1}$ in reaction buffer. Protein fractions were pooled and flash frozen.

### Synthesis of Phy-PP-GlcNAc$_2$

Phytol (Sigma Aldrich, W502200) was mixed with buffer (20 mM HEPES, pH 7.5; 150 mM NaCl; 1% DDM (w/v)) through sonication in an ultrasonic bath to form an opaque 10 mM stock suspension, which was used for the following steps. Phy-PP-GlcNAc$_2$ was synthesized using a phytol concentration of 100–500 μM phytol, a 5-fold molar excess of UDP-GlcNAc (Sigma) over phytol, and a 20-fold molar excess of ATP (Sigma) over phytol. *Sm*UdpK, *Sa*AglH, and *Sa*Agl24 were added at 50:1, 25:1 and 50:1 substrate to enzyme molar ratios, respectively. Reactions were carried out in buffer (20 mM HEPES, pH 7.5; 150 mM NaCl; 10 mM MgCl$_2$; 0.017 % DDM) and incubated at 45 °C overnight. Products were stored at −20 °C and used without further purification.

### Elongation of Phy-PP-GlcNAc$_2$ with cytoplasmic ALG mannosyltransferases

Synthesis of phytyl-PP-GlcNAc$_2$Man$_3$ was performed at RT overnight with 150–200 μM phytyl-PP-GlcNAc$_2$, a 5-fold molar excess of commercial GDP-Man (Sigma) over the LLO, and *Sc*ALG1 and *Ce*ALG2 added at a 100:1 and 20:1, respective substrate to enzyme molar ratios. Synthesis of phytyl-PP-GlcNAc$_2$Man$_5$ was performed as for phytyl-PP-GlcNAc$_2$Man$_3$ except that *Sc*ALG11 was included in a 100:1 molar substrate to enzyme ratio, and an eight-fold excess of GDP-Man over LLO was used. Reactions were performed in buffer (20 mM HEPES, pH 7.5; 150 mM NaCl; 10 mM MgCl$_2$ 0.017 % DDM; 0.0034% CHS; 5 mM β-mercaptoethanol) at 18 °C overnight. Products were incubated at 98 °C for 5 min to stop the enzymatic reactions and centrifuged at 17,500 × $g$ for 5 min at RT. The supernatant containing either phytyl-PP-GlcNAc$_2$Man$_3$ or phytyl-PP-GlcNAc$_2$Man$_5$ was stored at −20 °C until further use. For phytyl-PP-GlcNAc$_2$Man$_{6-9}$, no further purification was required, while for kinetic assays with *Tb*STT3A and *Tb*STT3B, solvent extraction was carried out as described in ref. 30. Briefly, the supernatant of the reaction mixture was flash-frozen and lyophilized overnight. The lyophilized solid was subjected to a 2:1 (v/v) CHCl$_3$:MeOH extraction, followed by a 10:10:3 (v/v/v) CHCl$_3$:MeOH:H$_2$O extraction step of the remaining pellet to recover phytyl-pyrophosphate-oligosaccharide (phytyl-PP-OS). This organic phase was dried under a nitrogen stream, lyophilized overnight and finally resuspended in 2 mM HEPES pH 7.4 buffer, before use with *Tb*STT3A and *Tb*STT3B.

### Synthesis of the donor substrates Phytyl-P-Man and Phytanyl-P-Glc

Phytylphosphate-mannose and phytanylphosphate-glucose were synthesized as follows. 100 μM of phytol (or phytanol) in 20 mM HEPES, pH 7.5, 150 mM NaCl, 1% DDM (w/v) was incubated with 2.5 μM UdpK, 500 μM ATP, 1 μM DPMS and 100 μM GDP-Man (or 500 uM GDP-Glc) overnight at 45 °C in donor synthesis reaction buffer (50 mM Tris/HCl pH 7.4, 200 mM NaCl, 10 mM MgCl$_2$, 0.03% DDM). After completion,

the reaction mixes were incubated at 98 °C for 5 min and centrifuged at $17,500 \times g$ for 5 min at RT to remove protein aggregates. The supernatant containing either phytylphosphate-mannose or phytanylphosphate-glucose was then directly used to provide the donor in reactions with ER-luminal mannosyltransferases or glucosyltransferases, respectively.

## Elongation of Phy-PP-GlcNAc$_2$Man$_5$ with ER-luminal ALG mannosyltransferases

Phytyl-PP-GlcNAc$_2$Man$_9$ production was carried out with 100 μM phytyl-PP-GlcNAc$_2$Man$_5$, a ten-fold molar excess of GDP-Man over LLO, $Sc$ALG3, $Hs$ALG9, and $Gg$ALG12, all at a 100:1 substrate to enzyme molar ratio. Synthesis and regeneration of the donor substrate was continuously carried out in the same reaction mix by the addition of 50 μM phytol, 1 mM ATP, ~1 μM $Pf$DPMS and 1-2 μM $Sm$UdpK. The reaction was performed at RT overnight in buffer (20 mM HEPES, pH 7.5; 150 mM NaCl; 0.017 % DDM; 0.0034 % CHS). After overnight incubation, the reaction was heated to 95 °C for 10 min before centrifuging for 5 min at $-16,000 \times g$. The supernatant was collected to remove the denatured enzymes, and the substrates were purified by solvent extraction as described above for phytyl-PP-GlcNAc$_2$Man$_{3-5}$.

## Elongation of Phy-PP-GlcNAc$_2$Man$_9$ with ER-luminal ALG glucosyltransferases

Phytyl-PP-GlcNAc$_2$Man$_9$Glc$_3$ production was carried out with 100 μM phytyl-PP-GlcNAc$_2$Man$_9$, a 15-fold molar excess of enzymatically synthesized phytanylphosphate-glucose over the LLO, and $Sc$ALG6, $Hs$ALG8, $Sc$ALG10, all at a 100:1 substrate to enzyme molar ratio. The reaction was performed at RT for 2 h in buffer (20 mM HEPES, pH 7.5; 150 mM NaCl; 0.017 % DDM; 0.0034 % CHS). Afterwards, the reaction was stopped by heating to 95 °C for 2 min and protein aggregates were removed by centrifuging the mixtures for 5 min at $-16,000 \times g$. The supernatant was collected and directly used to test transfer of glucosylated LLOs by the single-subunit oligosaccharyltransferases $Tb$STT3A and $Tb$STT3B. Phytyl-containing LLOs carrying only one or two glucoses on the A branch were produced as described above, with the exception that only $Sc$ALG6 was added to the reaction mix to afford phytyl-PP-GlcNAc$_2$Man$_9$Glc$_1$, while both $Sc$ALG6 and $Hs$ALG8 were provided to afford phytyl-PP-GlcNAc$_2$Man$_9$Glc$_2$.

## In vitro synthesis and validation of the synthesized Golgi-Man$_5$ glycan

Phytyl-PP-GlcNAc$_2$Man$_5$ displaying Golgi-Man$_5$ linkages was produced by extending Phy-PP-GlcNAc$_2$Man$_3$ with $Sc$ALG3 and $Gg$ALG12 in a 160:1 acceptor substrate to enzyme molar ratio. A 2.5-fold excess of Phy-P-Man donor was used, and the reaction was incubated for 2–4 h at room temperature in ER-luminal ALG mannosyltransferase buffer (see above). Phy-P-Man was synthesized using $Pf$DPMS as detailed above. The Golgi-Man$_5$ synthesized here was transferred to the peptide 5-FAM-GSDANYTYTQ-NH$_2$ using $Tb$STT3A prior to gel electrophoresis and visualization.

Human MGAT1 was used to extend the Golgi-Man$_5$ glycopeptide. Five μM glycopeptide was incubated with 200 nM $Hs$MGAT1 and 5 mM UDP-GlcNAc in reaction buffer (150 mM sodium chloride, 20 mM HEPES, pH 7.5, and 10 mM magnesium chloride) overnight at room temperature. Human MGAT1 was expressed and purified as previously published[71]. Digests of Golgi-Man$_5$ and ER-Man$_5$ glycopeptides were performed with an α1,2/3 mannosidase (New England Biolabs, P0729S). Reactions were carried out in a 20 μL reaction with 64 units of enzyme and 1 μM glycopeptide at 37 °C for 40 h in the buffer provided with the enzyme. Products of incubation with $Hs$MGAT1 or α1,2/3 mannosidase were separated via tricine gel electrophoresis and imaged using a fluorescent scanner.

## In vitro glycopeptide synthesis for monitoring $N$-glycan elongation by Agl and ALG enzymes

To confirm the correct assembly of the enzymatically generated phytyl-pyrophosphate-linked oligosaccharides reported above, we transferred the latter onto fluorescently labeled acceptor sequon using purified yeast oligosaccharyltransferase (OST) complex or the single-subunit OSTs A or B from $T.$ $brucei$. For yeast OST-catalyzed reaction, the peptide TAMRA-YANATS-NH$_2$ was used; for $Tb$STT3A-catalyzed reactions, 5-carboxyfluorescein-GSDANYTYTQ-NH$_2$ was used; finally, for $Tb$STT3B-catalyzed reactions, 5-carboxyfluorescein-GSLANYTK-NH$_2$ was used. Generally, peptide glycosylation was carried out at 30 °C with 5–10 μM fluorescent peptide, varying concentrations of the phytyl-PP-linked oligosaccharide to be tested, and 100-200 nM yeast OST complex, $Tb$STT3A or $Tb$STT3B, purified as described above. The reactions were performed in 150 mM NaCl, 20 mM HEPES, pH 7.5, 10 mM MnCl$_2$, 0.035% DDM, and 0.007% CHS. Glycopeptides were analyzed by SDS-PAGE and/or LC-MS/MS to assess the correct composition of the key glycan structures. For SDS-PAGE analysis, glycopeptides were diluted in Laemmli buffer, separated using Tricine-SDS-PAGE, and visualized using a fluorescence scanner. LC-MS/MS analysis ($n=1$ per glycopeptide) was performed at the Functional Genomics Center Zurich. Briefly, samples were diluted tenfold with 0.1% formic acid in H$_2$O, and 2 μL of the samples were analyzed on a calibrated FUSION mass spectrometer (Thermo Fischer Scientific) coupled to a nano-Acquity UPLC system (Waters). Glycopeptides were resuspended in 2.5% acetonitrile with 0.1% formic acid and loaded onto a nanoEase M/Z Symmetry C18, Trap Column (180 μm x 20 mm, 100 Å, 5 μm particle size) and separated on a nanoEase M/Z HSS C18 T3 (75 μm × 150 mm, 100 Å, 1.8 μm particle size), at a constant flow rate of 300 nL min-1, with a column temperature of 50 °C and a linear gradient of 2–35% acetonitrile/0.1% formic acid in 20 min, and then 35–98% acetonitrile/ 0.1% formic acid in 5 min, and then held isocratically for another 5 min. The mass spectrometer was operated under data-dependent acquisition (DDA), one scan cycle comprised of a full scan MS survey spectrum, followed by up to 12 sequential HCD MS/MS on the most intense signals above a threshold of 1e4. Full-scan MS spectra (600–2000 m/z) were acquired in the FT-Orbitrap at a resolution of 70,000 at 400 m/z, while HCD MS/MS spectra were recorded in the FT-Orbitrap at a resolution of 35,000 at 400 m/z. HCD was performed with a target value of 1e5, and a normalization collision energy 32 was applied. AGC target values were 5e5 for full FTMS. For all experiments, dynamic exclusion was used with a single repeat count, 15 s repeat duration, and 30 s exclusion duration. The XIC and MS profiles were shown by Xcalibur. All raw data were searched against a database containing user-provided sequences via Byonic 5.10 (Protein Metrics) search engine, with the consideration of amidation at the C-terminal and the corresponding modification at the N-terminal of either 5-CF or 5-TAMRA. The mass error for MS1 was 10 ppm and 0.02 Daltons for MS2.

## Kinetic analysis of peptide glycosylation using STT3 enzymes from *Trypanosoma brucei* and phytyl-pyrophosphate-oligosaccharides

In vitro glycosylation reactions were performed by mixing 50 nM purified $Tb$STT3A or 100 nM $Tb$STT3B protein, 15 μM phytyl-PP-linked oligosaccharide, 10 mM MnCl$_2$, 150 mM NaCl, 20 mM HEPES pH 7.5, 0.035% DDM, 0.007% CHS and 10 μM of the acceptor peptide (for STT3A: 5-FAM-GSDANYTYTQ-NH$_2$, for STT3B: 5-FAM-GSLANYTK-NH$_2$), and incubating at 30 °C. One microliter samples were taken at different time points and diluted in 10% acetonitrile in 10 mM K$_2$HPO4/ NaH$_2$PO$_4$, pH 7.0. Samples were analyzed by reverse-phase chromatography using a UPLC Dionex UltiMate 3000 with an Accucore 150-C18 100 × 2.1 mm 2.6 μm column (Thermo Fisher Scientific). Peaks for glycopeptide and peptide were integrated using the Software ChromeleonTM, and the amount of produced glycopeptide was determined for each data point.

## In vitro glycosylation of proteins containing three glycosylation sequons using *Tb*STT3A or *Tb*STT3B and phytyl-pyrophosphate-oligosaccharides

Protein glycosylation was carried out overnight at 30 °C with 2 μM Fluo-Trx-(A-sequon)$_3$ or Fluo-Trx-(B-sequon)$_3$, 0–40 μM LLO (displaying distinct mannose glycans), and 0.4 μM *Tb*STT3A or *Tb*STT3B. The reactions were performed in 150 mM NaCl, 20 mM HEPES, pH 7.5, 10 mM MnCl$_2$, 0.035% DDM, and 0.007% CHS. Glycosylation of the Fluo-Trx-(A-sequon)$_3$ construct was performed with *Tb*STT3A and Phy-PP-GlcNAc$_2$Man$_3$, Phy-PP-GlcNAc$_2$Man$_5$ or Phy-PP-GlcNAc$_2$Man$_9$, whereas the reactions with Fluo-Trx-(B-sequon)$_3$ were performed with *Tb*STT3B and Phy-PP-GlcNAc$_2$Man$_9$. Reacted samples were diluted in Laemmli buffer, separated using SDS-PAGE, and visualized using a fluorescence scanner.

## In vitro glycosylation of viral peptides with oligomannose glycans

All peptides were commercially synthesized (GenScript) with an N-terminal 5-carboxyfluorescein moiety followed by a Gly-Ser linker. Two versions of the HIV peptide representing the V3 variable region were used: V3-mini (sequence: 5-FAM-GSEINCTRPNNNTRPGEIIGD IRQAHCNISRAKWN-NH$_2$) and V3-mini-optimized (sequence: 5-FAM-GSEIQCTRDNNNTYPGEIIGDIRQADCNYTYAKWN-NH$_2$). The V3-mini peptide was based on a previously published peptide where the middle section of the V3 region was replaced with a Pro-Gly linker[62]. The V3-mini-optimized peptide was altered to increase its suitability as an acceptor substrate for *Tb*STT3A. The first glycosylation site (residue N297 of gp120) was abolished by mutation to glutamine, and the −2, +1, and +3 positions of the remaining sequons were mutated as indicated. The Dengue virus peptide was named NS1-N207 and had the sequence 5-FAM-GSIESSKNQTWQIEK-NH$_2$. For the Hepatitis C virus, two peptides coding for a version with enhanced solubility of the antigenic site 412 of the envelope glycoprotein E2 were designed, bearing the sequences: 5-FAM-GSRQLINTNGSWHIN-NH$_2$ (termed E2-N417); 5-FAM-GSRQLINTTG SWHIN-NH$_2$ (termed E2-N415).

Viral peptides were glycosylated overnight at 30 °C using 2–10 μM peptide, 150-300 nM *Tb*STT3A or *Tb*STT3B and up to 80 μM Phy-PP-GlcNAc$_2$Man$_3$, Phy-PP-GlcNAc$_2$Man$_5$ or Phy-PP-GlcNAc$_2$Man$_9$ in reaction buffer (150 mM NaCl, 20 mM HEPES, pH 7.5, 10 mM MnCl$_2$, 0.035% DDM, and 0.007% CHS). For the HIV peptides processed by *Tb*STT3A, 1 mM DTT was included in the buffer. The samples were heated briefly at 95 °C before tricine-SDS-PAGE[72] and imaging on a fluorescent scanner or MS analysis (see above).

## Figures preparation and data analysis

Structural figures were prepared using UCSF ChimeraX[73]. Protein sequence alignments were performed using Clustal Omega[74] and visualized using ESPript 3.0[75]. Graphs for enzyme kinetics were generated using GraphPad Prism 10. Densitometry was performed using ImageJ[76].

## Statistics and reproducibility

Protein expression, purification and enzymatic synthesis experiments have been carried out $n > 3$ times. Representative gels for protein and glycopeptide experiments are shown in Figs. 2a–d, 3a, 5b–d, 6c, f, g, j, k. For determination of kinetic parameters, a total of $n = 3$ replicates was run per condition.

## Reporting summary

Further information on research design is available in the Nature Portfolio Reporting Summary linked to this article.

## Data availability

Unless otherwise stated, all data supporting the results of this study can be found in the article, supplementary, and source data files. The mass spectrometry glyco-proteomics data generated in this study have been deposited in the GlycoPOST database[77] under accession code GPST000641 and to the ProteomeXchange Consortium via PRIDE[78] partner repository with the identifier PXD070840. The protein purification and activity data generated in this study are provided in the Supplementary Information. Source data are provided with this paper.

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

## Acknowledgements

The authors thank Meike Mikolin and Anna-Lena Schinke for technical assistance, and Dr. Chia-Wei Lin and the Functional Genomics Center Zurich (FGCZ) for mass spectrometry analysis. JANA is grateful for salary funding from the National Science and Engineering Research Council of Canada postdoctoral fellowship during this work. This work was supported by the Swiss National Science Foundation (grants 310030_196862 and 315230_220007 to K.P.L.).

## Author contributions

L.R., J.A.N.A., A.S.R. and K.P.L. conceived the project. L.R., J.A.N.A. and A.S.R. performed cloning, expression, and purification of the enzymes required for the pipeline, subsequent optimization of phytol-based LLOs synthesis, and carried out large-scale synthesis of phytol-based LLOs. L.R. and J.A.N.A. screened for different lipid substrates and determined initial conditions for one-pot reactions. A.S.R. performed kinetic assays of OST enzymes with phytol-based LLOs. J.A.N.A. performed cloning, expression, purification, fluorescent labeling, and in vitro glycosylation of multi-sequon-containing proteins. L.R. and J.A.N.A. performed viral peptides glycosylation. L.R., J.A.N.A., A.S.R. and K.P.L. wrote the manuscript.

## Competing interests

The authors have filed a patent application on the use of an enzymatic pipeline that employs some of the enzymes reported in the text (EP460037A1).
