## [Transparent Peer Review file · Nature Communications]

GLYCO-BUILD: An enzymatic pipeline for the synthesis of peptides carrying eukaryotic N-glycans

Corresponding Author: Dr Kaspar Locher

Version 0:

Reviewer comments:

Reviewer #1

(Remarks to the Author)

The article "GLYCO-BUILD: An enzymatic pipeline for the synthesis of high-mannose N-glycopeptides and N-glycoproteins" addresses a topic of high importance in basic and applied research, it is well written and to the best of my capacities, the English is correct. However, this current version for my consideration is not suited to be published in a journal such as Nat. communications. The results lack control datasets and proper validation. As an example, no measurement of the performance of the method described, since it could be of interest to potential users to know about synthetic efficiency. Enzyme kinetics analysis lacks statistical validation. Besides its potential relevance to the field, the results are shown in a very simplistic form, lacking in information for the readers, as earlier stated.

Reviewer #2

(Remarks to the Author)

Thank you for the opportunity to review this manuscript. We very much enjoyed reading about the work and acknowledge its importance and value for the glycobiology and bioprocessing fields.

The authors of this work developed an enzymatic pipeline mimicking the ER by synthesising GlcNAc2Man9 in near homogeneity and transfer on peptides with one or more glycosylation sites. They expressed a range of enzymes to efficiently perform each step. Some areas we feel need some attention:

1. The importance of the platform should be more clearly illustrated. The authors claim that their platform can be used to improve PK/PD of therapeutics. However, high mannose glycans are generally undesirable from a drug development viewpoint. Detailed discussion on how their platform can be used to change how therapeutics are produced is essential. Equally, the authors demonstrate they can efficiently produce vaccine targets but it is not appropriately explained as a potential platform. It would be good to include some details of the glycans typically present on viral surface proteins for the benefit of illustrating the potential uses of GLYCO-BUILD (e.g., what type of intermediate/immature structures?).
2. The inclusion of a brief review of in vitro enzymatic glycosylation methods and how the proposed approach is different would be useful.
3. The authors mention they could not achieve full conversion in all cases and suggest purification of desired structures. Do they have any thoughts on how reaction conditions could be optimised to achieve full conversion?
4. We enjoyed reading about the experiments where the authors modified multiple sites of proteins including of gp120 epitopes. Some of these epitopes have been engineered to facilitate enzyme activity. Have the authors confirmed that the final product would be equivalent in terms of functionality with the non-engineered epitope?
5. The authors present GLYCO-BUILD as a good platform for vaccine target production, so it would be interesting to see a binding assay and whether what they produced binds to bnAbs and can serve as a vaccine target.
6. A more balanced discussion that includes limitations of the platform is necessary, especially with respect to industrial applications, which are mentioned in the manuscript as the end goal.

Reviewer #3

(Remarks to the Author)

I co-reviewed this manuscript with one of the reviewers who provided the listed reports. This is part of the Nature

Communications initiative to facilitate training in peer review and to provide appropriate recognition for Early Career Researchers who co-review manuscripts.

Reviewer #4

(Remarks to the Author)

The authors of the manuscript entitled 'GLYCO-BUILD: An enzymatic pipeline for the synthesis of high-mannose N-glycopeptides and N-glycoproteins' have established a fully enzymatic workflow for the in vitro synthesis of several mannosylated N-glycans. By changing from dolichol-linked oligosaccharides to phytol-linked oligosaccharides the authors were able to complete the N-glycan processing steps that commonly occur in the endoplasmatic reticulum in eukaryotic cells and transfer these en bloc to glycopeptides. The authors claim that this novel pipeline facilitates the synthesis of biopharmaceutically interesting N-glycans that can be transferred to glycopeptides as well as glycoproteins. The manuscript is well written and in most cases is methodologically sound. However, I have some major concerns about the manuscript as listed below:

My major concern is that the presented manuscript has limited novelty. The authors present an elaborate enzymatic pipeline, but it seems that the majority of the enzymes have been used in previous studies. The in vitro synthesis of mannosylated glycan precursors (up to Man5) has been reported previously by Ramirez et al. (2017) and by Rexer et al (2020) (<https://doi.org/10.1016/j.jbiotec.2020.07.003>, which is not cited). In 2023, the latter authors extended their work by transferring more complex N-glycans (modified with Golgi-localized enzymes) to peptides (10.3389/fmolb.2023.1266431). The main novelties seem to be the adjustment from dolichol-linked oligosaccharides to phytol-linked oligosaccharides as lipid carriers and the synthesis of phytol-P-Man as substrate for later steps in the pipeline. In addition, the transfer of N-glycans to glycopeptides with STT3A from *Trypanosoma* is well established in literature.

Then, the claim that GLYCO-BUILD allows the synthesis of high-mannose N-glycans on glycoproteins is not supported by the presented data. The authors have only tested the transfer of their mannosylated N-glycans to glycopeptides, but not entire glycoproteins. The applicability of GLYCO-BUILD for for instance viral vaccines is described as they show promising results in eliciting desired immunological responses. But how can GLYCO-BUILD aid in mimicking the native glycosylation of entire viral antigens. Viral antigens are often highly glycosylated, but also differentially on different sequons. Some sequons are occupied with oligomannose N-glycans whereas other sequons are occupied with more complex N-glycans. I doubt whether it is possible to transfer multiple N-glycan species to a glycoprotein in a controlled manner using an in vitro strategy. Also, in higher eukaryotes, this transfer is catalyzed by the multi-subunit oligosaccharyltransferase (OST) complex and occurs co-translationally. I assume that in vitro glycosylation of glycoproteins only occurs on accessible sequons, whereas in vivo some sequons are glycosylated before/during the folding process and become less accessible after protein folding. So how can you control the glycosylation process?

Furthermore, the authors state in the introduction that 'the generation of high-mannose glycoconjugates' is a challenge, but I do not agree with this statement. The authors refer to overexpression in eukaryotic cells, such as CHO cells, and especially antibody production in these cells, which typically yields a heterogenous N-glycan composition. But antibodies are typically not the biopharmaceutical that carry oligomannose-type N-glycans. But more importantly, there are other options (besides CHO cells) to produce mannosylated N-glycans in expression systems: 1) insect cell-based expression and plant-based systems yield Man3 glycans, 2) yeast-based expression systems yield hypermannosylated N-glycans, but can be engineered to Man8 or Man5 glycans, 3) ER-retention signals (a 4 amino acid (KDEL fusion peptide) can be used to yield Man7/8/9 glycans on glycoproteins, or 4) the addition of kifunensine to the culture conditions of the used expression host can also yield Man7/8/9 glycans. Therefore, it seems quite an elaborate approach to generate typical mannosylated structures with an in vitro pipeline. And what about the associated costs of this enzymatic pipeline when applied on industrial scale. I expect the costs for a given biopharmaceutical to rise exponentially if it depends on a pipeline with at least 10 recombinant enzymes and expensive nucleotide sugar substrates. This latter aspect does not get attention in the manuscript.

Then the authors claim to have synthesized 'biotechnologically interesting glycans'. As far as I know, the majority of biopharmaceutical glycoproteins are decorated with more complex N-glycans than the structures that are synthesized in this study. But for those glycoproteins that receive immature glycans, they typically enter the Golgi-system carrying Man7/8 glycans and not Man9 structures. Since this Man9 structure is synthesized in a 1-pot reaction, it is impossible to generate homogenous Man8 structures in this way, but only in a sequential manner where the second enzymatic reaction with HsALG9 is skipped. In addition to the Man9 structure, I have also my doubts about the biotechnological relevance of the presented Man5 structure. The Man5 structure is synthesized with the α 1,2-mannosyltransferase ALG11, but if a typical secreted glycoprotein would carry Man5 structures it would not carry a α 1,2-linked mannose at all. This Man5 structure only carries a α 1,3- and a α 1,6-linked mannoses after the action of Golgi-resident mannosidases. So, these are two completely different structures.

The authors envision great applicability in terms of studying cellular N-glycosylation and the generation of glycosylated diagnostic and therapeutic agents. The authors don't provide examples for how GLYCO-BUILD could aid investigation into cellular glycosylation and I have difficulty envisioning this myself. I fully agree that custom-made glycopeptides with tailored glycans can aid in the development of diagnostics, but about therapeutic approaches I am skeptical still. As mentioned, the presented study does not glycosylate entire glycoproteins and as illustrated above, I foresee major obstacles for custom-made in vitro glycosylation of biopharmaceutical glycoproteins. The production of defined viral glycopeptides can for sure aid in diagnostic approaches, but in terms of vaccination I am doubting whether a glycopeptide is sufficient to elicit a competent antibody response. The glycopeptide might indeed be recognized by specific B cells, but maturation of these cells (and possibly class switching to desired isotypes) relies on signals coming from CD4 T helper cells. These cells can

best be stimulated with entire proteins instead of small peptides. The authors also state that the GLYCO-BUILD pipeline may improve pharmacokinetic and pharmacodynamic properties of biopharmaceuticals, such as half-life. However, the investigated mannosylated glycans are typically associated with reduced half-lives by serum clearance via mannose-binding C-type lectin receptors. Furthermore, I am wondering how stable the recombinant enzymes are (not discussed in the manuscript)? Can they be stored for longer periods without losing activity? Or can the synthesized lipid-linked glycan structures be stored for longer periods. Lack of stability would greatly impact applicability as well.

All combined, I think that the presented manuscript does not present major breakthroughs in the field (as I would expect for a journal as Nature Communications) and that the applicability of the pipeline that is claimed is not completely supported by the presented data. Therefore, I would recommend rejecting the manuscript for publication.

Additional comments:

The authors have not confirmed the different linkages of the mannose residues upon in vitro synthesis.

Why do the authors use different peptides for glycosylation with the yeast OST complex and TbSTT3A? This is not explained.

The authors also describe that they have performed different techniques for assessing the in vitro glycosylation of different peptides with yeast OST or TbSTT3A. But, from the text it is not immediately clear that LC-MS analysis has not been performed for the TbSTT3A reactions, but only gel electrophoretic analysis. Can the authors elaborate on their choices?

What does 'near quantitative glycosylation' mean? Or is 'near complete glycosylation' meant? Later on, the term 'quantitative glycosylation' is used, whereas the detection method by gel is not used in a quantitative way.

For STT3B a new peptide was developed, but the enzyme was not able to fully glycosylate it. What about STT3A? Was that tested as well? Would this problem be solved by simply using STT3A?

Afterwards, the authors state that transfer of the GlcNAc2Man3 and GlcNAc2Man5 were too low, but the data are not shown. And this is not a surprise based on figure 4.

Figure 3: The legend is confusing as the masses indicated here do not correspond to masses in the MS profiles.

The description "g0, g1, and g2 shown on the right refer to unglycosylated peptide or peptide featuring single or double glycosylation" is missing in figure 5 and extended figure 3.

Version 1:

Reviewer comments:

Reviewer #1

(Remarks to the Author)

Thanks to the authors for clarifying and for raising my observations.

Reviewer #2

(Remarks to the Author)

Our previous comments have been addressed.

Reviewer #3

(Remarks to the Author)

Reviewer #4

(Remarks to the Author)

The authors of the manuscript entitled 'GLYCO-BUILD: An enzymatic pipeline for the synthesis of peptides carrying eukaryotic N-glycans' have gone at length to prepare their revision. I have to admit that I am not an expert in the field of in

vitro glyco-engineering and I am not fully up to date with all the literature in this discipline. The authors present a compelling response letter to address the comments/issues of all 4 reviewers, and where necessary present the appropriate literature to back up their arguments. The manuscript has been revised thoroughly, and I have to complement the authors for their work. The manuscript reads very well and is better balanced. The novelty is now presented more clearly as well. Furthermore, I really appreciate the addition of the new data on the synthesis of glucosylated glycans, but more importantly the Golgi-Man5 glycans and extension with HsMGAT1.

In summary, the authors have done an excellent job in addressing the reviewer comments. In its current form the manuscript is well suited for publication.

Responses to the Reviewers

We thank the editor and the reviewers for their feedback, which helped strengthen our manuscript. Below are our responses (in blue). We have implemented extensive changes throughout the text. Whenever possible, we indicated below the precise edits or their location within the text.

Reviewer #1 (Remarks to the Author):

The article "GLYCO-BUILD: An enzymatic pipeline for the synthesis of high-mannose N-glycopeptides and N-glycoproteins" addresses a topic of high importance in basic and applied research, it is well written and to the best of my capacities, the English is correct.

We thank the reviewer for the evaluation and for stating the importance of the topic.

However, this current version for my consideration is not suited to be published in a journal such as Nat. communications. The results lack control datasets and proper validation. As an example, no measurement of the performance of the method described, since it could be of interest to potential users to know about synthetic efficiency.

We have modified to text to improve clarity and more clearly indicate the synthetic efficiency of each step or pot of our pipeline. On page 7, we added: "The reaction yield for Pot 1 and all subsequent steps reached >95% product formation under the conditions reported in the Methods section, unless specified otherwise". Statements specific to each reaction Pot are reported at the end of paragraphs in pages 8, 9, 10. For clarity, we replaced "quantitative glycosylation" with "complete glycosylation" when describing reactions yielding full conversion with no discernible side products.

Enzyme kinetics analysis lacks statistical validation.

We think the reviewer might have missed the legend of Figure 4 and Extended Data Figure 5 (updated numbering), where we specified that each data point was measured in triplicate and the standard error was also included. We modified Figure 4 and Extended Data Figure 5 and their caption to show the individual data points instead of the mean values for better visualization.

Besides its potential relevance to the field, the results are shown in a very simplistic form, lacking in information for the readers, as earlier stated.

We have restructured our manuscript and included additional efficiency data and methodological information. In addition, we have included additional data showing that the pipeline can be used to generate glucosylated and hybrid glycans, further extending the scope of the work. Please refer to the new paragraphs in pages 9, 10, 22, 23, 29, 30 (main text); updated Figures 1, 2, 3 (Figures file); updated Ext. Data Fig. 1; new Ext. Data Fig. 2, 3 and 4 (Extended Data).

Reviewer #2 (Remarks to the Author):

Thank you for the opportunity to review this manuscript. We very much enjoyed reading about the work and acknowledge its importance and value for the glycobiology and bioprocessing fields. The authors of this work developed an enzymatic pipeline mimicking the ER by synthesising GlcNAc2Man9 in near homogeneity and transfer on peptides with one or more glycosylation sites. They expressed a range of enzymes to efficiently perform each step.

We thank the reviewer(s) for the kind evaluation and for appreciating our work.

Some areas we feel need some attention:

1. The importance of the platform should be more clearly illustrated. The authors claim that their platform can be used to improve PK/PD of therapeutics. However, high mannose glycans are generally

undesirable from a drug development viewpoint. Detailed discussion on how their platform can be used to change how therapeutics are produced is essential.

We agree with the reviewer(s) that high mannose glycans (i.e. Man₅-Man₉) are generally not useful to improve PK/PD of therapeutics, while complex *N*-glycans have multiple demonstrated effects in this regard. We have modified the text to emphasize that our pipeline can produce precursors of glycopeptides bearing complex and hybrid glycans. Please refer to the edit in the Discussion paragraph: "Second, the GlcNAc₂Man₃ and Golgi-Man₅ glycans generated using our pipeline can serve as starting points for generating complex and hybrid glycopeptides by enzymatic methods developed previously^{28,64,65}. This might help improve the pharmacokinetics and pharmacodynamics of peptides and proteins used as biologics, for example by modulating their half-life^{28,64}". It is these precursors that constituted a barrier in the past. We envision that the GlcNAc₂Man₃ glycan on our glycopeptides can be extended with GlcNAc, Gal and Sia moieties by soluble Golgi glycosyltransferase as previously published (PMIDs: 36280670, 38321209; [https://www.cell.com/chem/abstract/S2451-9294\(24\)00227-4](https://www.cell.com/chem/abstract/S2451-9294(24)00227-4)). This would give rise to a (poly)peptide modified with biantennary complex *N*-glycans, with the latter having the potential to improve the half-life of therapeutics. We have updated Figures 1, 2 and added Extended Data Figures 2 and 3 to better convey our results.

Equally, the authors demonstrate they can efficiently produce vaccine targets but it is not appropriately explained as a potential platform. It would be good to include some details of the glycans typically present on viral surface proteins for the benefit of illustrating the potential uses of GLYCO-BUILD (e.g., what type of intermediate/immature structures?).

We have amended the text to add more context on the application of our pipeline. Edit on page 12: "Short viral glycosylated antigens (glycopeptides) that are able to induce relevant B cells and T cell responses might find applications in diagnostics or immunization studies¹⁴⁻¹⁶. We therefore tested whether our pipeline facilitated the generation of viral glycopeptide epitopes known to carry oligomannose glycans". Edit on page 15: "We envision that the generation of viral glycopeptides with defined and homogeneous oligomannose glycans could benefit both serum testing and immunological investigations, as such glycoconjugates might recapitulate aspects of the humoral immune response elicited by fully folded viral surface proteins¹⁶". We have included reference showing that glycopeptides including their glycans are recognised by T-cell receptors and are essential for efficient B-cell activation (PMID: 32439962).

2. The inclusion of a brief review of in vitro enzymatic glycosylation methods and how the proposed approach is different would be useful.

We have restructured the manuscript. Please see page 4 of the main text.

3. The authors mention they could not achieve full conversion in all cases and suggest purification of desired structures. Do they have any thoughts on how reaction conditions could be optimised to achieve full conversion?

Glycan transfer is affected by the spacing between sequons, optimal amino acid sequence, and glycan size (Man₅ can be completely transferred, Man₉ currently less than completely). In the future, engineered variants of the OST enzymes might increase catalytic activity.

4. We enjoyed reading about the experiments where the authors modified multiple sites of proteins including of gp120 epitopes. Some of these epitopes have been engineered to facilitate enzyme activity. Have the authors confirmed that the final product would be equivalent in terms of functionality with the non-engineered epitope?

We are pursuing such studies in collaboration. However, this manuscript describes the design of engineered epitope variants that could be efficiently glycosylated using our pipeline. We therefore feel that further characterization of the produced glycopeptides was out of the scope of this manuscript.

5. The authors present GLYCO-BUILD as a good platform for vaccine target production, so it would be interesting to see a binding assay and whether what they produced binds to bnAbs and can serve as a vaccine target.

The ability to produce glycosylated peptides could lead to new vaccines (PMIDs: 32439962, 29107699, 28298421). However, the production of vaccines is out of the scope of this manuscript. We have therefore removed this statement from page 14 (~~"We envision that the production of viral glycopeptides with specific high mannose glycan structures could have applications both in serum testing and vaccine development"~~) and have instead indicated that this technology could be developed for the production of glycosylated peptides in the future (page 12: "We therefore tested whether our pipeline facilitated the generation of viral glycopeptide epitopes known to carry oligomannose glycans").

6. A more balanced discussion that includes limitations of the platform is necessary, especially with respect to industrial applications, which are mentioned in the manuscript as the end goal.

We agree with the reviewer and have incorporated such discussion into the manuscript. We added on page 14: "The current purification yields of functional enzymes fully support the needs for academic research labs ... We anticipate that further optimization and scale-up of our workflow will enable its transition to a higher-scale setting".

Reviewer #3 (Remarks to the Author):

We thank the reviewer for the joint evaluation and the suggestions to improve our manuscript.

Reviewer #4 (Remarks to the Author):

The authors of the manuscript entitled 'GLYCO-BUILD: An enzymatic pipeline for the synthesis of high-mannose N-glycopeptides and N-glycoproteins' have established a fully enzymatic workflow for the in vitro synthesis of several mannosylated N-glycans. By changing from dolichol-linked oligosaccharides to phytol-linked oligosaccharides the authors were able to complete the N-glycan processing steps that commonly occur in the endoplasmic reticulum in eukaryotic cells and transfer these en bloc to glycopeptides. The authors claim that this novel pipeline facilitates the synthesis of biopharmaceutically interesting N-glycans that can be transferred to glycopeptides as well as glycoproteins. The manuscript is well written and in most cases is methodologically sound.

We thank the reviewer for the evaluation and the comments on our work. We have addressed all of them (see below) and included them in our revised manuscript.

However, I have some major concerns about the manuscript as listed below:

My major concern is that the presented manuscript has limited novelty. The authors present an elaborate enzymatic pipeline, but it seems that the majority of the enzymes have been used in previous studies. The in vitro synthesis of mannosylated glycan precursors (up to Man5) has been reported previously by Ramirez et al. (2017) and by Rexer et al (2020) (<https://doi.org/10.1016/j.jbiotec.2020.07.003>, which is not cited). In 2023, the latter authors extended their work by transferring more complex N-glycans (modified with Golgi-localized enzymes) to peptides (10.3389/fmolb.2023.1266431). The main novelties seem to be the adjustment from dolichol-linked oligosaccharides to phytol-linked oligosaccharides as lipid carriers and the synthesis of phytol-P-Man as substrate for later steps in the pipeline.

There appears to be some misunderstanding or misconception. Our study contains substantial novelty with respect to published literature as listed in the following:

- We report for the first time **a set of recombinant enzymes and new reaction conditions that can efficiently generate lipid-PP-linked-(GlcNAc)₂ at the needed scale.** This advance overcomes a main challenge with respect to glycan donor availability (as mentioned in Wenzel et al., 2023). We achieve this by avoiding chemical synthesis, which is prohibitive for most biochemistry labs. Indeed, in Rexer et al. (2020) (now cited, thank you for pointing it out) and Wenzel et al. (2023), the authors relied on a contract with an external company to produce their lipid-PP-linked-(GlcNAc)₂. With our pipeline, we eliminate this issue and improve accessibility of these moieties to academic labs, a breakthrough for the field of *in vitro* glycoengineering.
- **The enzymes or their expression and purification conditions (and constructs) are distinct compared to previous literature.** Indeed, the advances made possible by GLYCO-BUILD relied on extensive enzyme homolog screening and optimization of all necessary steps. The enzymes mentioned in the literature cannot accomplish what our study can. Specifically:
 - (i) these following homologs have been reported here for the first time: ALG2 from *Caenorhabditis elegans*, ALG12 from *Gallus gallus*, ALG9 from *Homo sapiens*, STT3B from *Trypanosoma brucei*. Switching to these homologs was essential for achieving the results we describe.
 - (ii) New fusion constructs or protocol for expression-purification and functional activity for the following enzymes: *SmUdpK*, *SaAglH*, *SaAgl24*, *ScALG1*, *ScALG11*, *HsALG8* and *ScALG10*. Our work substantially improved purity and activity of the enzymes.
 - (iii) Repurposing of enzymes for *in vitro* synthesis: to our knowledge, there is no report in literature of the use of *SmUdpK* in lipid-phosphate production for *N*-glycan synthesis, nor are there published reports of recombinant over expression and purification of *SaAglH*. While enzymatic assays for *SaAgl24* have been reported, the use of this enzyme for the *in vitro* generation of *N*-glycans has not been reported.
- **Synthetic efficiency, purity and scale:** Rexer et al. (2020) report a synthetic efficacy of ~50% after 26.5 h of reaction time and the product corresponds to a heterogeneous mixture of Man₄- and Man₅-LLO. Furthermore, a very minor fraction of the acceptor peptide was modified with glycans by *TbSTT3A*. Similar results were obtained in Wenzel et al. (2023). In contrast, our study results in close to 100% yield (formation of homogeneous LLOs and glycopeptides, no side products detectable), as judged by densitometry of fluorescent glycopeptides or LC-MS/MS. In addition, the work from Ramírez et al. (2017) relied on chemically synthesized lipid-linked chitobiose, which limited scale, applicability and accessibility of the approach. With GLYCO-BUILD, the use of phytol (inexpensive off-the-shelf compound) and optimized recombinant enzymes democratize *N*-glycan *in vitro* synthesis and transfer.
- After the initial submission of our paper we have also discovered **a shortcut to producing hybrid glycans** and have included new data in Figures 1, 2 as well as Extended Data Figures 2 and 3. Our work now allows easy production of homogeneous Golgi-Man₅ precursor that is needed for the generation of custom hybrid glycopeptides. We believe that the ability to produce hybrid glycans will have academic as well as biotechnological uses. Bespoke hybrid glycopeptides could be used to study the effect of particular glycans on their immunogenicity, effect on PK/PD, etc.
- After the initial submission of our paper we have also developed **a method to generate glucosylated *N*-glycans and LLOs based on phytol**, compounds that are not accessible otherwise from natural sources and that are useful for the investigation of protein folding and cellular glycosylation. Also, we generated lipid-P-glucose, the sugar donor for the study and use of diverse glucosyltransferases. This new findings have been included in Figures 1, 2, 3, and Extended Data Figures 1 and 3.

In addition, the transfer of *N*-glycans to glycopeptides with STT3A from *Trypanosoma* is well established in literature.

We are familiar with the use of *TbSTT3A* in the synthesis of glycopeptides *in vitro*, given that the study was produced in our lab. The study is also referenced in our paper. However, these 2017 results relied on synthetic dolichol analogs and did not report any experiments using phytol as a lipid carrier. In contrast, we are now using phytol. In addition, we include *TbSTT3B* as an alternative for glycosylation sequons that are not surrounded by acidic environments and are therefore not efficiently glycosylated by *TbSTT3A*. We are also providing a novel strategy for the glycosylation of multiple sequons and investigate the kinetics of the transfer of different glycan structures.

Then, the claim that GLYCO-BUILD allows the synthesis of high-mannose N-glycans on glycoproteins is not supported by the presented data. The authors have only tested the transfer of their mannosylated N-glycans to glycopeptides, but not entire glycoproteins.

We think that the reviewer might have missed the data presented in Figure 5 and Extended Data Figure 6, where we demonstrated transfer of mannosylated N-glycans to the flexible C-terminus of glycoproteins. We would like to clarify that these reactions were performed with full-length protein. However, we realize that this glycosylation is performed on engineered unstructured regions and have added on page 11: "We next explored whether we could glycosylate up to three closely spaced glycosylation sequons fused to the terminus of a folded protein".

The applicability of GLYCO-BUILD for for instance viral vaccines is described as they show promising results in eliciting desired immunological responses. But how can GLYCO-BUILD aid in mimicking the native glycosylation of entire viral antigens. Viral antigens are often highly glycosylated, but also differentially on different sequons. Some sequons are occupied with oligomannose N-glycans whereas other sequons are occupied with more complex N-glycans. I doubt whether it is possible to transfer multiple N-glycan species to a glycoprotein in a controlled manner using an in vitro strategy. Also, in higher eukaryotes, this transfer is catalyzed by the multi-subunit oligosaccharyltransferase (OST) complex and occurs co-translationally. I assume that in vitro glycosylation of glycoproteins only occurs on accessible sequons, whereas in vivo some sequons are glycosylated before/during the folding process and become less accessible after protein folding. So how can you control the glycosylation process?

We want to clarify that we did not state in our manuscript that we want to glycosylate folded viral proteins, which would be challenging, except when modifying exposed loops. In contrast, we focused our attention on viral glycopeptides. There is an increasing body of literature that supports the notion that even short viral glycosylated antigens (glycopeptides) are able to induce relevant B cells and T cell responses. This especially applies to selected epitopes, such as HIV V3 mini-epitope or a glycopeptide derived from the V2 region of HIV gp120. In this context, GLYCO-BUILD can be used to produce those antigens at scale and in a homogeneous manner for diagnostics or immunization studies. Selected references on the topic:

1. Cai, H. *et al.* Synthetic Three-Component HIV-1 V3 Glycopeptide Immunogens Induce Glycan-Dependent Antibody Responses. *Cell Chem Biol* **24**, 1513-1522 e1514 (2017). <https://doi.org/10.1016/j.chembiol.2017.09.005>
2. Alam, S. M. *et al.* Mimicry of an HIV broadly neutralizing antibody epitope with a synthetic glycopeptide. *Sci Transl Med* **9** (2017). <https://doi.org/10.1126/scitranslmed.aai7521>
3. Sun, L. *et al.* Glycopeptide epitope facilitates HIV-1 envelope specific humoral immune responses by eliciting T cell help. *Nat. Commun.* **11**, 2550 (2020).

Furthermore, the authors state in the introduction that 'the generation of high-mannose glycoconjugates' is a challenge, but I do not agree with this statement. The authors refer to overexpression in eukaryotic cells, such as CHO cells, and especially antibody production in these cells, which typically yields a heterogenous N-glycan composition. But antibodies are typically not the biopharmaceutical that carry oligomannose-type N-glycans. But more importantly, there are other options (besides CHO cells) to produce mannosylated N-glycans in expression systems: 1) insect cell-based expression and plant-based systems yield Man3 glycans, 2) yeast-based expression systems yield hypermannosylated N-glycans, but can be engineered to Man8 or Man5 glycans, 3) ER-retention signals (a 4 amino acid (KDEL fusion peptide) can be used to yield Man7/8/9 glycans on glycoproteins, or 4) the addition of kifunensine to the culture conditions of the used expression host can also yield Man7/8/9 glycans. Therefore, it seems quite an elaborate approach to generate typical mannosylated structures with an in vitro pipeline.

The systems mentioned by the reviewer are known to result in macro- and micro-heterogeneity and be subjected to batch-to-batch variations. We designed GLYCO-BUILD for improving glycan homogeneity for applications where heterogeneity is undesirable, as is the case with developing certain biopharmaceuticals. We have replaced the sentence "A specific challenge is the generation of high-mannose glycoconjugates, as they cannot be enriched or purified in large amounts from natural sources" with "Glycoconjugates representing intermediates or end products of the ER glycosylation

pathways cannot be enriched or purified in large amounts from natural sources” (page 3 of the main text).

And what about the associated costs of this enzymatic pipeline when applied on industrial scale. I expect the costs for a given biopharmaceutical to rise exponentially if it depends on a pipeline with at least 10 recombinant enzymes and expensive nucleotide sugar substrates. This latter aspect does not get attention in the manuscript.

We amended the text to give a more balanced view on the biotechnological applications of GLYCO-BUILD at higher scale. We agree that the present study does not represent a business case but a method for generating highly homogeneous *N*-glycopeptides using only enzymes, not synthetic chemistry. We added on page 14: “The current purification yields of functional enzymes fully support the needs for academic research labs ... We anticipate that further optimization and scale-up of our workflow will enable its transition to a higher-scale setting”.

Then the authors claim to have synthesized ‘biotechnologically interesting glycans’. As far as I know, the majority of biopharmaceutical glycoproteins are decorated with more complex *N*-glycans than the structures that are synthesized in this study. But for those glycoproteins that receive immature glycans, they typically enter the Golgi-system carrying Man_{7/8} glycans and not Man₉ structures. Since this Man₉ structure is synthesized in a 1-pot reaction, it is impossible to generate homogenous Man₈ structures in this way, but only in a sequential manner where the second enzymatic reaction with HsALG9 is skipped.

We want to clarify that GLYCO-BUILD can be used to synthesize a repertoire of glycan structures. Please refer to updated Fig. 1. We can also synthesize a homogeneous Man₈ structure using a two-step reaction approach which we have highlighted in the text (page 8): “We could synthesize the oligomannose intermediates Phy-PP-GlcNAc₂Man₆, Phy-PP-GlcNAc₂Man₇, and Phy-PP-GlcNAc₂Man₈, depending on the ALG enzymes included in the reaction mixture (Fig. 2a)”. Our pipeline also includes the production of Man₃ *N*-glycans which can be used as precursor for the synthesis of the complex glycan structures suggested by the reviewer or the Golgi Man₅ precursor required for the production of hybrid *N*-glycans. Please refer to new paragraphs in pages 9 and 30 (main text); updated Figures 1, 2 (Figures file); new Ext. Data Fig. 2 and 3 (Extended Data).

In addition to the Man₉ structure, I have also my doubts about the biotechnological relevance of the presented Man₅ structure. The Man₅ structure is synthesized with the α 1,2-mannosyltransferase ALG11, but if a typical secreted glycoprotein would carry Man₅ structures it would not carry α 1,2-linked mannose at all. This Man₅ structure only carries α 1,3- and α 1,6-linked mannoses after the action of Golgi-resident mannosidases. So, these are two completely different structures.

We clarified in the text when we are referring to the ER Man₅ structure. Additionally, we have new experimental data detailing a fully enzymatic method of synthesizing Golgi-Man₅ glycan referred to above. As mentioned, we have discovered a shortcut to producing hybrid glycans and have included new experimental data in Figures 1, 2 as well as Extended Data Figures 2 and 3. Our work allows easy production of homogeneous Golgi-Man₅ precursor that is needed for the generation of custom hybrid glycopeptides.

The authors envision great applicability in terms of studying cellular *N*-glycosylation and the generation of glycosylated diagnostic and therapeutic agents. The authors don’t provide examples for how GLYCO-BUILD could aid investigation into cellular glycosylation and I have difficulty envisioning this myself.

We respectfully disagree with the reviewer. We and others have investigated the structure and catalytic mechanism of several enzymes in the cellular glycosylation machinery. Those studies have strongly benefited from having access to substrate homologs for trapping different states of their catalytic cycle (e.g. PMIDs: 29058712, 29386647, 32103179, 36435935, 36604564). In the case of *N*-glycosylation, the availability of such compounds has been limited due to the requirement of highly complex chemical synthesis. GLYCO-BUILD provides an entirely enzymatic approach to produce such molecules, which is accessible to any protein biochemistry lab. We envision that this approach will allow the glycobiology

community to synthesize phytol-linked oligosaccharides, phytol-linked glucosylated *N*-glycans, phytol-phosphate mannose or glucose and glycopeptides that could help the investigation of enzymes involved in the calnexin/calreticulin and UDP-Glc:Glycoprotein Glucosyltransferase (UGGT)-based folding processes, Golgi glycosyltransferases involved in *N*-glycan remodelling, mannosyltransferases (C-mannosylation, O-mannosylation, GPI-anchor biosynthesis), among others. We have modified pages 14 and 15 (Discussion) accordingly.

I fully agree that custom-made glycopeptides with tailored glycans can aid in the development of diagnostics, but about therapeutic approaches I am skeptical still. As mentioned, the presented study does not glycosylate entire glycoproteins and as illustrated above, I foresee major obstacles for custom-made in vitro glycosylation of biopharmaceutical glycoproteins. The production of defined viral glycopeptides can for sure aid in diagnostic approaches, but in terms of vaccination I am doubting whether a glycopeptide is sufficient to elicit a competent antibody response.

Our manuscript focuses on the efficient production of highly homogeneous glycopeptides. Therefore, a thorough characterization of these molecules' potential applications is outside of the scope of this study and we look forward to other groups to test these compounds. In addition, a consistent body of literature has already shown that certain glycopeptides are sufficient to elicit antibody responses in animal studies. Please refer to:

1. Cai, H. *et al.* Synthetic Three-Component HIV-1 V3 Glycopeptide Immunogens Induce Glycan-Dependent Antibody Responses. *Cell Chem Biol* **24**, 1513-1522 e1514 (2017). <https://doi.org/10.1016/j.chembiol.2017.09.005>
2. Alam, S. M. *et al.* Mimicry of an HIV broadly neutralizing antibody epitope with a synthetic glycopeptide. *Sci Transl Med* **9** (2017). <https://doi.org/10.1126/scitranslmed.aai7521>

The glycopeptide might indeed be recognized by specific B cells, but maturation of these cells (and possibly class switching to desired isotypes) relies on signals coming from CD4 T helper cells. These cells can best be stimulated with entire proteins instead of small peptides.

There are examples of short glycopeptides that can activate CD4+T cells. For example: Sun, L. *et al.* "Glycopeptide epitope facilitates HIV-1 envelope specific humoral immune responses by eliciting T cell help". *Nat. Commun.* **11**, 2550 (2020). This is the reference 16 of the paper.

The authors also state that the GLYCO-BUILD pipeline may improve pharmacokinetic and pharmacodynamic properties of biopharmaceuticals, such as half-life. However, the investigated mannosylated glycans are typically associated with reduced half-lives by serum clearance via mannose-binding C-type lectin receptors.

We are aware that high-mannose *N*-glycans are associated with reduced half-lives and we do not plan to use them directly, to improve the PK of therapeutics. However, GLYCO-BUILD can produce peptides modified with GlcNAc₂Man₃ or Golgi-Man₅ glycans. These moieties represent the core of complex and hybrid *N*-glycans, respectively, and this can be further extended by commonly used Golgi glycosyltransferases to build up complex *N*-glycans, such as sialylated ones, that can improve the serum half-life (PMIDs: 36280670, 38321209; [https://www.cell.com/chem/abstract/S2451-9294\(24\)00227-4](https://www.cell.com/chem/abstract/S2451-9294(24)00227-4)). Many peptide-based diagnostics or therapeutics would strongly benefit from improved physicochemical properties and longer serum half-life which could be modulated with glycosylation (PMIDs 28660018, 20826563). Please refer to the edit in the Discussion paragraph: "Second, the GlcNAc₂Man₃ and Golgi-Man₅ glycans generated using our pipeline can serve as starting points for generating complex and hybrid glycopeptides by enzymatic methods developed previously^{28,64,65}. This might help improve the pharmacokinetics and pharmacodynamics of peptides and proteins used as biologics, for example by modulating their half-life^{28,64}".

Furthermore, I am wondering how stable the recombinant enzymes are (not discussed in the manuscript)? Can they be stored for longer periods without losing activity? Or can the synthesized lipid-linked glycan structures be stored for longer periods. Lack of stability would greatly impact applicability as well.

We tested these conditions and included the relevant information in the manuscript. We added on page 6: "All proteins used in the pipeline described below displayed good stability and could be frozen, allowing long-term storage for future use". In brief, stability of LLOs or enzymes stored at -80°C is preserved.

All combined, I think that the presented manuscript does not present major breakthroughs in the field (as I would expect for a journal as Nature Communications) and that the applicability of the pipeline that is claimed is not completely supported by the presented data. Therefore, I would recommend rejecting the manuscript for publication.

We think that the new version of the manuscript presents a clearer explanation of the novelty of our work and how it stands out from published literature. Additionally, our new experimental data allowing the production of Golgi-Man₅ and glucosylated *N*-glycans opens entirely new areas of research for study. We also clarified that we were able to glycosylate acceptor sequons while displayed on glycoproteins, and that our work mostly focuses on GLYCO-BUILD as a highly efficient pipeline for production of glycopeptides. Among the many uses we envision for GLYCO-BUILD is the production of viral glycosylated antigens, which is grounded on a consistent body of literature that supports the use of peptidic glyco-epitopes for research and translational applications. Finally, we included new experiments, detailed above, and addressed all points raised by the reviewer.

Additional comments:

The authors have not confirmed the different linkages of the mannose residues upon *in vitro* synthesis.

All the enzymes used in the pipeline display exquisite regio- and stereo-selectivity. Furthermore, we have extensively characterized the correct linkage formation of our ER-LLO analogs *in vitro* by treatment with specific mannosidases combined with MS analysis (PMIDs 28575298, 32103179, 36435935). For this manuscript, we have included such an analysis for the validation of the ER- and Golgi-Man₅ glycan structures. This data is presented now in Extended Data Figure 3.

Why do the authors use different peptides for glycosylation with the yeast OST complex and TbSTT3A? This is not explained.

Both peptides come from independent optimization for each OST system:

TbSTT3A PMID: 28204532

Yeast OST PMID: 34023382

The authors also describe that they have performed different techniques for assessing the *in vitro* glycosylation of different peptides with yeast OST or TbSTT3A. But, from the text it is not immediately clear that LC-MS analysis has not been performed for the TbSTT3A reactions, but only gel electrophoretic analysis. Can the authors elaborate on their choices?

For kinetic measurements of TbSTT3A and TbSTT3B, we used LC (C18 RP column and fluorescence detector) as the substrate peptides for *in vitro* measurements are fluorescently labeled. To validate glycan elongation, without kinetic measurements, we run the glycopeptides on tricine gels. At certain key steps of the pipeline, selected glycan moieties were transferred by yeast OST onto a different fluorescent peptide and analysed by MS, to check quality (homogeneity) and mass of the glycan. *In vitro* glycosylation of proteins was followed by separation on SDS-PAGE. Glycosylation of our carrier protein was followed by observing the retarded mobility of the protein with each glycosylated sequon.

What does 'near quantitative glycosylation' mean? Or is 'near complete glycosylation' meant? Later on, the term 'quantitative glycosylation' is used, whereas the detection method by gel is not used in a quantitative way.

Yes, for “near quantitative glycosylation” we meant “near complete glycosylation”. We modified this throughout the entire text.

For STT3B a new peptide was developed, but the enzyme was not able to fully glycosylate it. What about STT3A? Was that tested as well? Would this problem be solved by simply using STT3A?

No, *Tb*STT3A is not as efficient as *Tb*STT3B in glycosylating the acceptor sequon GSLANYTK.

Afterwards, the authors state that transfer of the GlcNAc₂Man₃ and GlcNAc₂Man₅ were too low, but the data are not shown. And this is not a surprise based on figure 4.

Yes, *Tb*STT3B displays strong specificity for the Man₉ glycan donor. Please refer to Figure 4.

Figure 3: The legend is confusing as the masses indicated here do not correspond to masses in the MS profiles.

We modified the legend of Figure 3 to improve clarity.

The description “g0, g1, and g2 shown on the right refer to unglycosylated peptide or peptide featuring single or double glycosylation” is missing in figure 5 and extended figure 3.

We added the missing text into the corresponding figure captions.

We thank the reviewers for their positive response and constructive, helpful suggestions to improve the manuscript throughout the review process. No further changes in response to reviewer comments were required.

REVIEWERS' COMMENTS

Reviewer #1 (Remarks to the Author):

Thanks to the authors for clarifying and for raising my observations.

Reviewer #2 (Remarks to the Author):

Our previous comments have been addressed.

Reviewer #3 (Remarks to the Author):

Reviewer #4 (Remarks to the Author):

The authors of the manuscript entitled 'GLYCO-BUILD: An enzymatic pipeline for the synthesis of peptides carrying eukaryotic N-glycans' have gone at length to prepare their revision. I have to admit that I am not an expert in the field of in vitro glyco-engineering and I am not fully up to date with all the literature in this discipline. The authors present a compelling response letter to address the comments/issues of all 4 reviewers, and where necessary present the appropriate literature to back up their arguments. The manuscript has been revised thoroughly, and I have to complement the authors for their work. The manuscript reads very well and is better balanced. The novelty is now presented more clearly as well. Furthermore, I really appreciate the addition of the new data on the synthesis of glucosylated glycans, but more importantly the Golgi-Man5 glycans and extension with HsMGAT1.

In summary, the authors have done an excellent job in addressing the reviewer comments. In its current form the manuscript is well suited for publication.